# A widespread hydrogenase supports fermentative growth of gut bacteria in healthy people

Caitlin Welsh [1,2,3] ✉, Princess R. Cabotaje[4], Vanessa R. Marcelino [5,6], Thomas D. Watts [1], Duncan J. Kountz [7], Marion Jespersen[1], Jodee A. Gould[2], Nhu Quynh Doan[1], James P. Lingford[1], Thilini Koralegedara[1], Jessica Solari[1], Gemma L. D'Adamo[2,3], Ping Huang [4], Natasha Bong[1], Emily L. Gulliver [2,3], Remy B. Young[2,3], Henrik Land [4], Kaija Walter [4], Isaac Cann [8], Gabriel V. Pereira [9], Eric C. Martens [9], Patricia G. Wolf[8,10], Jason M. Ridlon [8], H. Rex Gaskins [8], Edward M. Giles [2,11], Dena Lyras [1], Rachael Lappan [1], Gustav Berggren [4], Samuel C. Forster [2,3] ✉ & Chris Greening [1] ✉

Disruption of hydrogen ($H_2$) cycling in the gut is linked to gastrointestinal disorders, infections and cancers. However, the mechanisms and microorganisms controlling $H_2$ production in the gut remain unresolved. Here we show that gut $H_2$ production is primarily driven by the microbial group B [FeFe]-hydrogenase. Metagenomics and metatranscriptomics of stool and tissue biopsy samples show that hydrogenase-encoding genes are widely present and transcribed in gut bacteria. Assessment of 19 taxonomically diverse gut isolates revealed that the group B [FeFe]-hydrogenases produce large amounts of $H_2$ gas and support fermentative growth of Bacteroidetes and Firmicutes. Further biochemical and spectroscopic characterization of purified enzymes show that they are catalytically active, bind a di-iron active site and reoxidize ferredoxin derived from the pyruvate:ferredoxin oxidoreductase reaction. Group B hydrogenase-encoding genes are significantly depleted in favour of other fermentative hydrogenases in patients with Crohn's disease. Finally, metabolically flexible respiratory bacteria may be the dominant hydrogenotrophs in the gut, rather than acetogens, methanogens and sulfate reducers. These results uncover the enzymes and microorganisms controlling $H_2$ cycling in the healthy human gut.

Molecular hydrogen ($H_2$) is a central intermediate in gastrointestinal digestive processes. Most bacteria within the gut hydrolyse and ferment dietary carbohydrates to absorbable short-chain fatty acids[1–3] and large quantities of $H_2$ gas[4,5]. $H_2$ accumulates to high micromolar levels in the gut, where it is primarily consumed by other microorganisms for energy conservation and carbon fixation[6,7], although a large proportion is also expelled as flatus or exhaled[8–10]. Classically, three groups of gut microorganisms are thought to consume $H_2$, namely, acetogenic bacteria, methanogenic archaea and sulfate-reducing bacteria[3,11–14]. $H_2$ consumption by gut microorganisms lowers $H_2$ partial pressures, thereby ensuring fermentation remains thermodynamically favourable[3,15–19]. In turn, many $H_2$-producing and $H_2$-consuming microorganisms form mutualistic relationships by conducting interspecies $H_2$ transfer dependent on physical association[15,20]. In addition to supporting digestion, gastrointestinal $H_2$ cycling modulates levels of important metabolites in the gut, including butyrate[21], hydrogen sulfide[22], bile

acids[2] and host steroids[23], with diverse effects on processes such as digestion, inflammation and carcinogenesis. It is also proposed that microbiota-derived or therapeutically supplied $H_2$ may directly benefit human cells as an anti-oxidant[24,25]. Disruption of the balance between $H_2$-producing and $H_2$-consuming bacteria has been linked to a range of gut and wider disorders[19,26]; most notably, gas build-up contributes to the symptoms of irritable bowel syndrome and hydrogen breath tests are frequently, if controversially, used to detect disorders such as carbohydrate malabsorption[27–29]. Numerous pathogens also exploit microbiota-derived $H_2$ during invasion, including *Helicobacter pylori* and *Salmonella*[30–34], or rapidly produce it in the case of pathogenic Clostridia and protists[30,35,36].

Despite the central importance of $H_2$ cycling in human health and disease, surprisingly little is known about which microorganisms and enzymes mediate this process. Both the production and consumption of $H_2$ are catalysed by hydrogenases, which fall into three major groups dependent on the metal content of their active sites: the [FeFe]-, [NiFe]-, and [Fe]-hydrogenases, with multiple subgroups[37,38]. It has been classically thought that most $H_2$ production in the gut is mediated by fermentative bacteria, primarily the class Clostridia, that couple reoxidation of ferredoxin (for example, reduced during acetate fermentation by the pyruvate:ferredoxin oxidoreductase (PFOR) reaction) to the evolution of $H_2$. Some Clostridia use group A1 [FeFe]-hydrogenases, an extensively structurally and mechanistically characterized lineage of enzymes, to rapidly produce $H_2$ (refs. [39,40]). Some $H_2$ may also be produced by formate hydrogenlyase complexes (containing a group 4a [NiFe]-hydrogenase) that disproportionate formate during fermentative survival of Enterobacteriaceae[41,42]. Yet two recent findings suggest that other fermenters are also active in the human gut. Our 2016 survey showed genes encoding a distantly related enzyme called the group B [FeFe]-hydrogenase (28% amino acid identity to the group A1 enzymes) are widespread in diverse gut isolates and abundant in gut metagenomes[19,43]. Two recent biochemical studies[44,45] suggest that these enzymes are active and predominantly mediate $H_2$ production, although their physiological activity and role has yet to be confirmed in any organism. In parallel, electron-confurcating hydrogenase complexes (group A3 [FeFe]-hydrogenases) have been discovered that couple oxidation of NAD(P)H and reduced ferredoxin to the evolution of $H_2$ (refs. [46–49]). We have shown that group A3 [FeFe]-hydrogenases are primarily responsible for $H_2$ production in ruminants[17,50,51], although it is unclear if these principles also extend to humans. Similarly, it is unclear whether the paradigms regarding $H_2$ consumption are accurate, given the three classical groups of $H_2$ oxidizers (hydrogenotrophs) are generally in low abundance in the human gut. Indeed, only approximately half of people produce methane gas[52,53], and it is becoming increasingly apparent that most hydrogen sulfide is derived from organosulfur compounds rather than sulfate reduction[54–56]. Respiratory hydrogenotrophs that use electron acceptors such as fumarate, nitrate, sulfoxides and inflammation-derived oxygen may also be active but overlooked members of gut microbiota[19,31].

Here we have integrated enzymatic and microbial insights to build a detailed picture of $H_2$ cycling in the human gastrointestinal tract. To do so, we holistically profiled the abundance, transcription and distribution of hydrogenases using metagenomes and metatranscriptomes, including original biopsy samples, in both healthy individuals and those with gastrointestinal disorders. We then performed an in-depth analysis of 19 bacterial isolates and 4 heterologously produced hydrogenases to confirm the activity and roles of these enzymes. We reveal that group B [FeFe]-hydrogenase drives most $H_2$ production in the human gut, highlight the overlooked role of *Bacteroides* as major $H_2$-producing fermenters, and show that hydrogenase genes are differentially abundant between healthy people and those with chronic disease phenotypes, such as Crohn's disease.

## Results

### Group B [FeFe]-hydrogenase genes are widespread and transcribed in the human gut

We initially investigated the distribution of hydrogenase genes in the human gut by analysing 300 human stool metagenomes[57] (Supplementary Dataset 1). Hydrogenase genes are extremely abundant, occurring on average at $1.44 \pm 0.58$ copies per genome (cpg) (Fig. 1a). By far the most abundant are the functionally uncharacterized group B [FeFe]-hydrogenases ($0.75 \pm 0.25$ cpg), hypothesized but unproven to mediate fermentative $H_2$ production[19] (Fig. 1b). Genes encoding these enzymes are much more abundant than the ferredoxin-dependent group A1 [FeFe]-hydrogenases ($0.10 \pm 0.09$ cpg), which were previously thought to account for most gut $H_2$ production[6,58,59], and electron-confurcating group A3 [FeFe]-hydrogenases ($0.19 \pm 0.11$ cpg) that dominate $H_2$ production in ruminants[17,50]. Other enzymes also potentially play minor roles in $H_2$ production in the human gut, including formate hydrogenlyases (group 4a [NiFe]-hydrogenase genes; $0.02 \pm 0.07$ cpg) and possibly ferredoxin-dependent energy-converting hydrogenases (group 4e [NiFe]-hydrogenase genes; $0.06 \pm 0.04$ cpg) (Fig. 1a,b and Supplementary Dataset 1). Consistently, analyses of 78 paired metatranscriptomes confirm that these hydrogenase genes are highly transcribed (RNA:DNA ratios between 1.76 and 6.90, depending on subgroup) (Supplementary Dataset 1 and Fig. 1a). Transcripts for the group B [FeFe]-hydrogenase genes are the most numerous ($95 \pm 86$ reads per kilobase of transcript per million mapped reads (RPKM); RNA:DNA ratio = 2.0) and 3.3-fold, 4.7-fold and 26-fold higher than the well-characterized group A3, A1 and 4a enzymes (Fig. 1a). Given [FeFe]-hydrogenases are usually highly active enzymes[39,40], transcription of these levels probably enables rapid $H_2$ production in the gut. There was nevertheless much interindividual variation in expression levels. Genes encoding nitrogenases, which produce $H_2$ during their reaction cycle[60], were also widely encoded but minimally transcribed by gut bacteria (Fig. 1a). Altogether, group B [FeFe]-hydrogenases probably drive most $H_2$ production in the gut but operate alongside other $H_2$-producing hydrogenases.

To infer which gut microorganisms encode the genes for these enzymes, we mapped the hydrogenase-encoding reads to both our comprehensive hydrogenase database (HydDB)[61] and our in-house collection of 812 sequenced gut isolates (Supplementary Dataset 2). Genes encoding the group B [FeFe]-hydrogenases are very widespread among gut bacteria, encoded by 62% of isolates and the dominant gut phyla Firmicutes, Bacteroidetes and Actinobacteria (Supplementary Dataset 2 and Fig. 2). The abundant lineage *Bacteroides* (Supplementary Dataset 3) accounted for the most group B [FeFe]-hydrogenase reads in the metagenomes and metatranscriptomes, followed by *Alistipes* and Clostridia lineages such as *Faecalibacterium*, *Agathobacter* and *Roseburia* (Fig. 1c and Supplementary Dataset 1). The group A1 and A3 [FeFe]-hydrogenase genes were also widespread, encoded and transcribed by various Bacteroidia, Clostridia and Fusobacteria, whereas formate hydrogenlyases were restricted to Enterobacteriaceae, Pasteurellaceae and Coriobacteriaceae (Figs. 1c and 2 and Supplementary Dataset 1).

A small but active proportion of the community is predicted to mediate $H_2$ uptake in the human gut. Genes for group 1 [NiFe]-hydrogenases, which support anaerobic respiration using electron acceptors such as fumarate, nitrate, nitrite and sulfite[37,38], are encoded by 9% of gut bacteria based on metagenomic short reads (Supplementary Dataset 1 and Fig. 1a,b) and 6% of our isolates (Supplementary Dataset 2 and Fig. 2). As evidenced by the extremely high standard deviations of their metagenome counts ($0.09 \pm 0.17$ cpg) and metatranscriptome reads ($51 \pm 153$ RPKM), the abundance and transcription of these genes greatly vary between individuals (Fig. 1a). They were primarily encoded and transcribed by Enterobacteriaceae ([NiFe] group 1c and 1d), which are known to use gut-derived $H_2$ as a respiratory energy source during colonization[30,62,63], and by lineages such

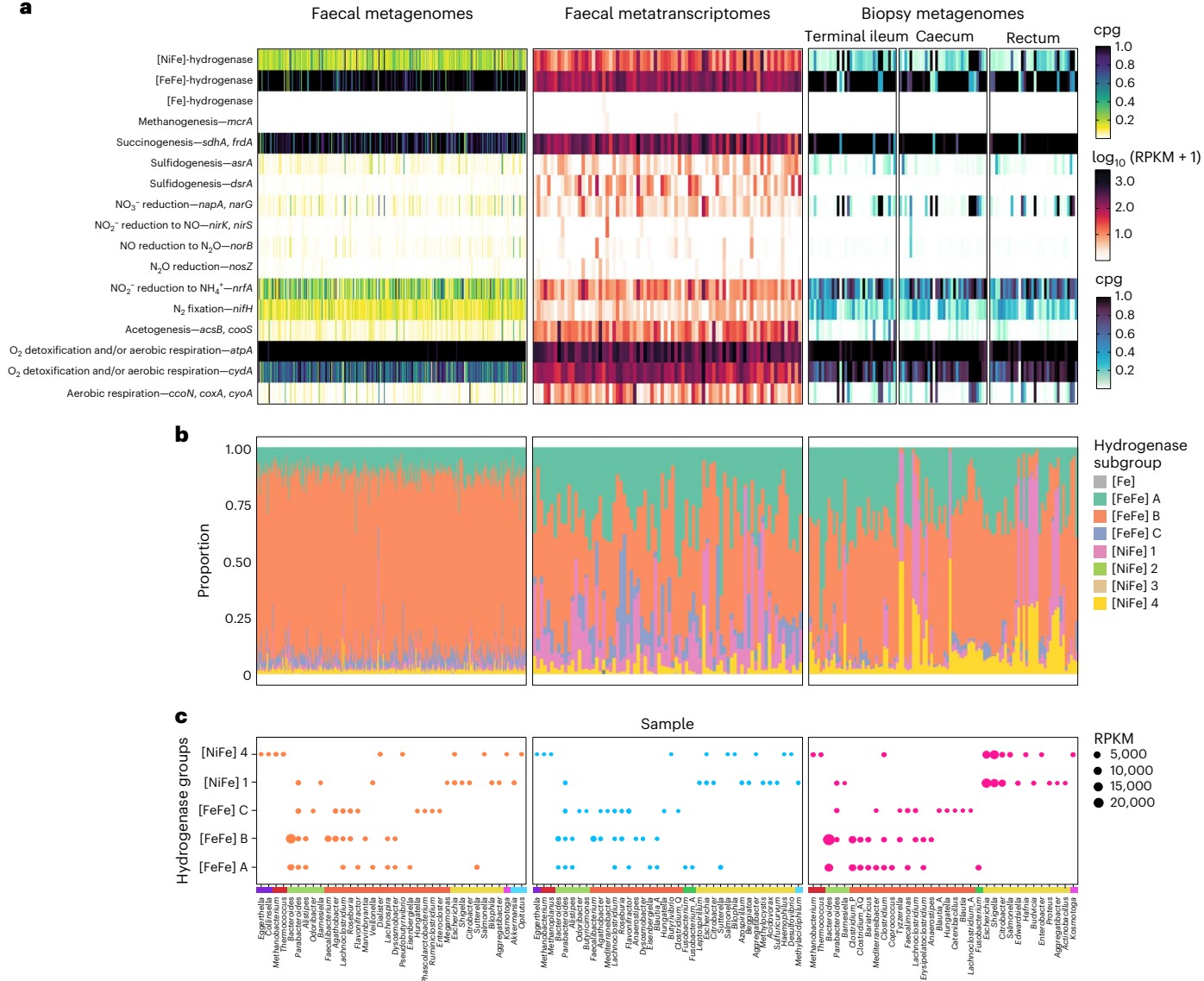

**Fig. 1 | Abundance, transcription and distribution of hydrogenase genes and H₂-related metabolic genes throughout the human gut. a**, Abundance and transcription of the genes encoding the catalytic subunits of the three types of hydrogenases and the terminal reductases known to use H₂-derived electrons in faecal metagenomes (left; $n = 300$), faecal metatranscriptomes (middle; $n = 78$) and biopsy enrichment metagenomes (right; $n = 102$). These results summarize homology-based searches against comprehensive reference databases and are shown in average gene copies per organism (normalized to a set of universal single-copy ribosomal genes) for metagenomes and RPKM for metatranscriptomes. **b**, Proportion of each hydrogenase group present in each sample per dataset. **c**, Top genera predicted to encode or transcribe hydrogenases for each dataset. The top ten most abundant genera are included for the five most abundant gut hydrogenase lineages, expressed in RPKM. Phyla are represented by the coloured bar above the genera names. Purple represents Actinobacteria; red, Archaea; light green, Bacteroidetes; orange, Firmicutes; dark green, Fusobacteria; yellow, Proteobacteria; pink, Thermotogae; and light blue, Verrucomicrobia.

as *Veillonella* (1d), *Parabacteroides* (1d) and *Akkermansia* (1f), whose H₂ metabolism remains to be investigated (Fig. 1c). Some group A3 [FeFe]-hydrogenase genes were also encoded by hydrogenotrophic acetogens such as *Blautia*, where these enzymes oxidize H₂, rather than produce it, in contrast to fermenters[64]. Genes encoding the group 3 and 4 [NiFe]-hydrogenases and [Fe]-hydrogenases of methanogenic archaea were also detected in a subset of samples. Consistently, we also detected genes encoding the signature enzymes responsible for fumarate, sulfite, nitrate, and nitrite reduction, acetogenesis, and methanogenesis in the metagenomes and metatranscriptomes (Fig. 1a and Supplementary Dataset 1). Although these genes were in low abundance (except those encoding fumarate reductase), they were often highly transcribed (RNA:DNA ratios of 54 for acetyl-coenzyme A (CoA) synthase, 37 for dissimilatory sulfite reductase and 4 for respiratory

nitrate reductase) (Supplementary Dataset 1). Phylogenomic analysis of the gut isolates also revealed frequent co-occurrence of genes encoding group 1 [NiFe]-hydrogenases with respiratory reductases (Fig. 2). However, it should be noted that the respiratory reductases can accept electrons from a range of both organic and inorganic donors other than H₂. Also detected were genes encoding putative sensory hydrogenases (group C [FeFe]-hydrogenases, $0.11 \pm 0.15$ cpg) (Fig. 1a), which are thought to differentially regulate [FeFe]-hydrogenases in response to H₂ accumulation in Clostridia and probably other lineages[17,37,49].

We tested whether these findings also extend to microbiota sampled within gut tissues, given stool samples provide a biased assessment of gut microbial content[65–67]. To do so, we collected mucosal biopsies from the terminal ileum, caecum and rectum of 42 donors, and then enriched and sequenced their microbiota[68] (Supplementary

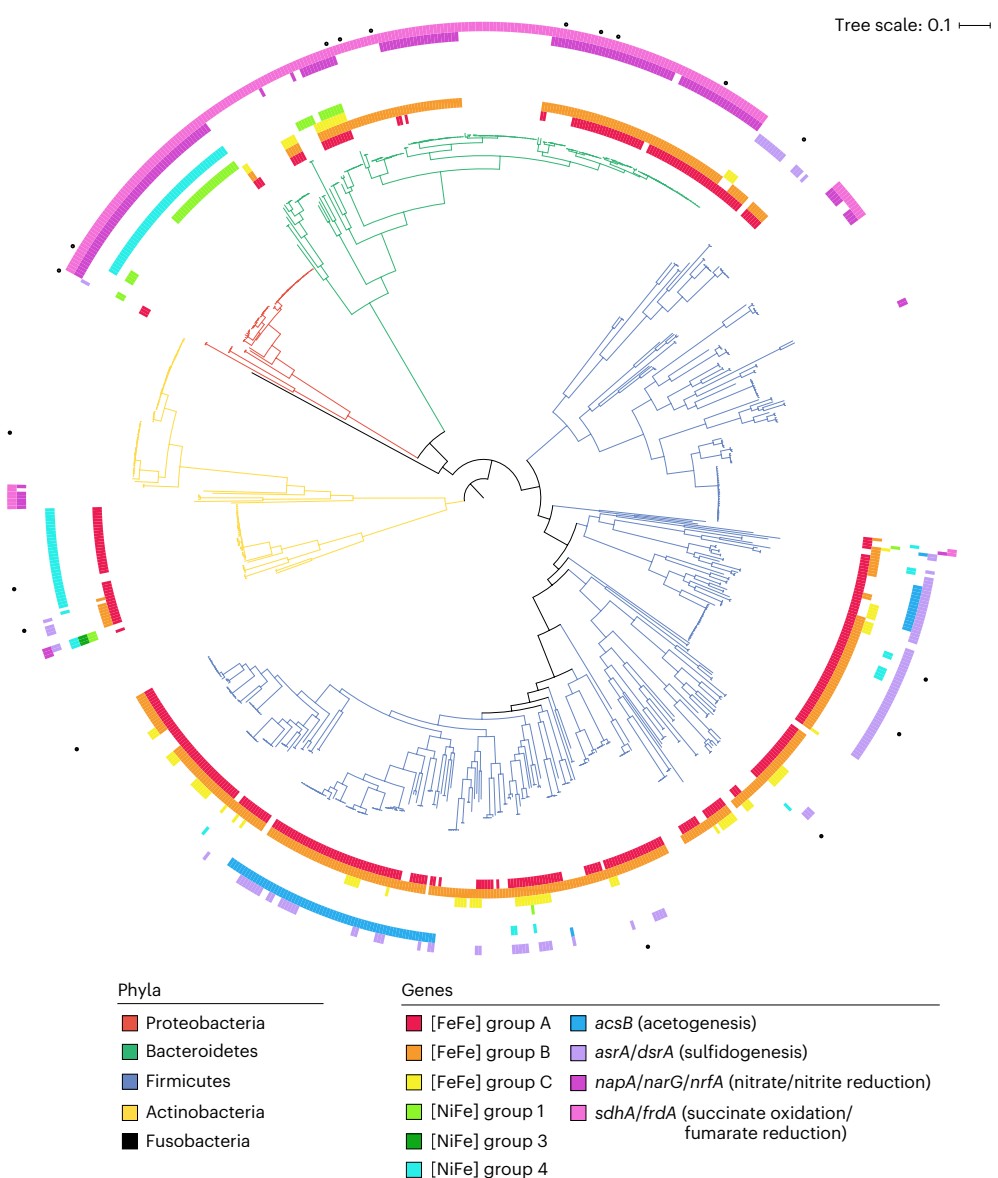

Tree scale: 0.1 ⊢——⊣

**Phyla**

- 🟥 Proteobacteria
- 🟩 Bacteroidetes
- 🟦 Firmicutes
- 🟨 Actinobacteria
- ⬛ Fusobacteria

**Genes**

- 🟥 [FeFe] group A
- 🟧 [FeFe] group B
- 🟨 [FeFe] group C
- 🟩 [NiFe] group 1
- 🟩 [NiFe] group 3
- 🟦 [NiFe] group 4
- 🟦 *acsB* (acetogenesis)
- 🟪 *asrA/dsrA* (sulfidogenesis)
- 🟪 *napA/narG/nrfA* (nitrate/nitrite reduction)
- 🟪 *sdhA/frdA* (succinate oxidation/ fumarate reduction)

**Fig. 2 | Phylogenomic tree showing distribution of hydrogenase genes among 812 bacterial isolates from the human gut.** Isolates are from the five dominant phyla within the human gut, with branch colours showing their phylum-level taxonomy. Isolates were shown to encode the catalytic subunit genes coding for the major groups of gut hydrogenases and the terminal reductases associated with methanogenesis, acetogenesis, sulfidogenesis, nitrate reduction or succinogenesis (coloured rings). The tree was generated using approximately maximum-likelihood estimation with the Jukes–Cantor model (via FastTree) and standardized 'bac120' phylogenetic analysis (via GTDB-Tk) and was midpoint rooted. Results are based on homology-based searches against comprehensive reference databases. Specific isolates were selected for further analysis, including culture-based activity measurements and transcriptome studies (black dots). The tree scale represents the branch length of the tree, as calculated by the number of base substitutions per base position.

Dataset 1). Concordantly, group B [FeFe]-hydrogenases were by far the most abundant hydrogenase genes detected across these mucosal biopsy samples (0.75 ± 0.25 cpg); they were 3.7-fold more abundant than the next most abundant hydrogenase genes (group A3 [FeFe]-hydrogenases) and primarily encoded by *Bacteroides* based on read mapping (Fig. 1). The group 1c, 1 d and 4a [NiFe]-hydrogenase genes were also enriched by 6.1-fold, 2.6-fold and 7.0-fold in the biopsy compared with stool metagenomes; this probably reflects the adherence of Enterobacteriaceae to the gut luminal walls, where they potentially use microbiota-derived $H_2$ to support anaerobic and potentially even aerobic respiration (Supplementary Dataset 1 and Fig. 1b,c). Thus, the group B [FeFe]-hydrogenase probably contributes substantially to fermentative $H_2$ production throughout the intestines, much of which is possibly recycled by respiratory hydrogenotrophs. No significant differences in hydrogenase content were found between intestinal regions, which was probably masked by the high degree of interindividual variation.

## Group B [FeFe]-hydrogenases are expressed and active in diverse gut isolates

To confirm whether the group B [FeFe]-hydrogenase is active, we used gas chromatography to test $H_2$ production of 19 phylogenetically and physiologically diverse bacterial gut isolates each grown on standard yeast casitone fatty acids (YCFA) medium under fermentative conditions (Supplementary Table 1 and Supplementary Fig. 1). Of these isolates, 13 encoded group B [FeFe]-hydrogenase genes, either individually or together with other hydrogenases, all but 1 of which produced high levels of $H_2$ (Fig. 3a and Supplementary Fig. 2). This collection included 7 *Bacteroides* isolates that each rapidly produced headspace $H_2$ to average maximum levels of 3.0 ± 0.6% during fermentative growth, and 4 genera

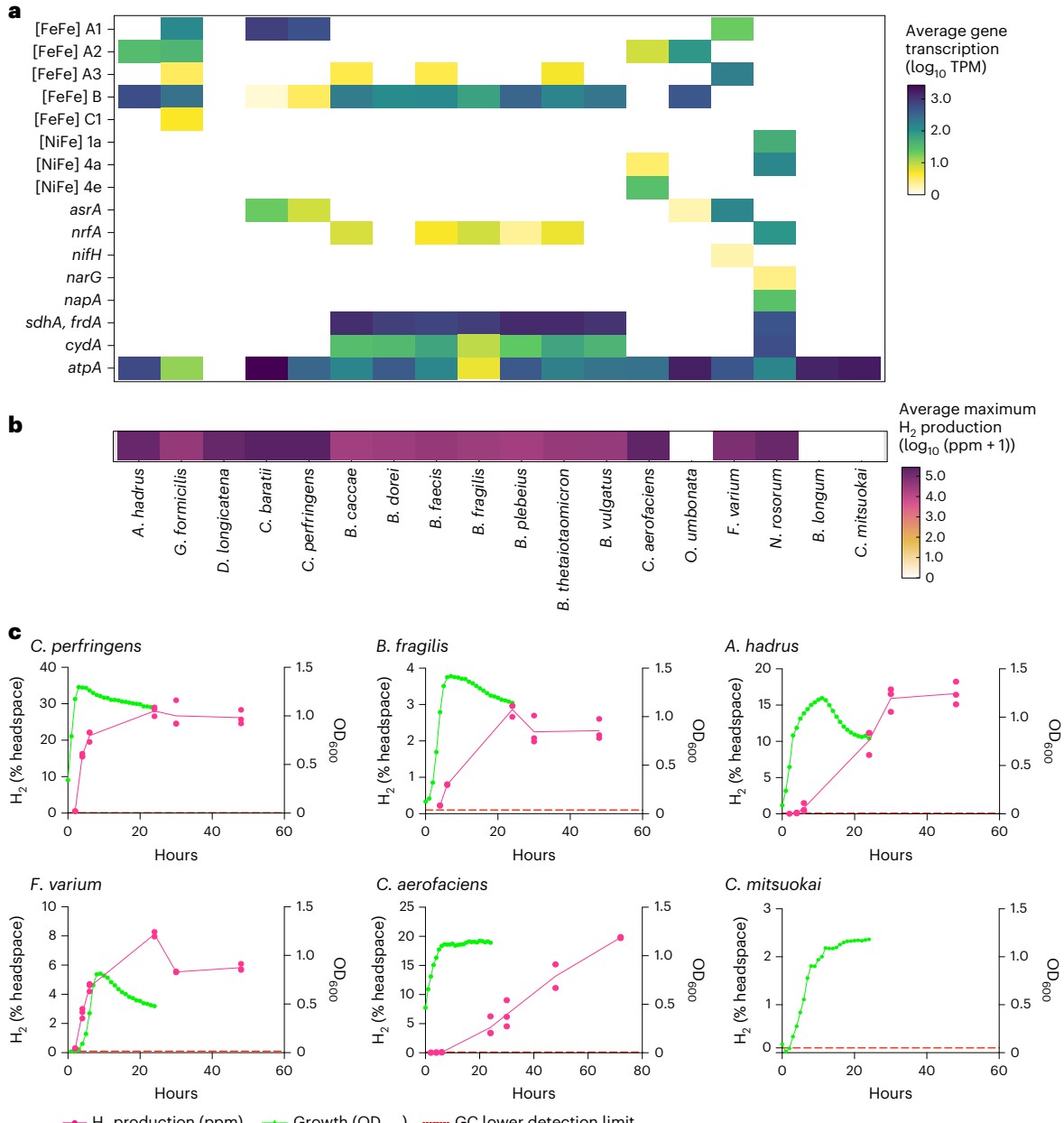

**Fig. 3 | Hydrogenase transcription and activity across 18 bacterial gut isolates. a**, A heat map showing the average transcription levels (expressed as $\log_{10}$ TPM) of the catalytic subunit genes for hydrogenases and the terminal reductases associated with sulfidogenesis, succinogenesis, nitrate reduction and aerobic respiration. **b**, A heat map showing the average maximum $H_2$ production for each isolate (expressed as $\log_{10}$ (ppm + 1)). In both heat maps, results show means from biologically independent triplicates. *Bifidobacterium longum* and *Catenibacterium mitsuokai* do not encode hydrogenase genes and

so are used as negative controls. **c**, Bacterial growth measured by $OD_{600}$ (green lines) and $H_2$ production (% of headspace; pink lines) of representative isolates from chosen phyla over 24–72 hour periods ($n = 3$), where the lower detection threshold of the gas chromatograph is 1,000 ppm (dashed red line). *A. hadrus*, *Anaerostipes hadrus*; *B. dorei*, *Bacteroides dorei*; *C. aerofaciens*, *Collinsella aerofaciens*; *C. baratii*, *Clostridium baratii*; *D. longicatena*, *Dorea longicatena*; *F. varium*, *Fusobacterium varium*; *G. formicilis*, *Gemmiger formicilis*; *N. rosorum*, *Necropsobacter rosorum*; *O. umbonata*, *Olsenella umbonata*.

from the class Clostridia (Fig. 3b, Supplementary Fig. 1 and Supplementary Table 1). We compared these activities with those of six positive and negative control isolates (Fig. 3a and Supplementary Fig. 1): no $H_2$ was detected in the three isolates lacking hydrogenases; high levels of $H_2$ were produced during fermentative growth of a *Fusobacterium* isolate with a prototypical group A1 [FeFe]-hydrogenase gene; and $H_2$ was produced during stationary phase in bacteria encoding formate hydrogenlyases, in line with their confirmed roles[42,69]. Altogether, these analyses show that $H_2$ production is a widespread trait among gut bacteria that encode group B [FeFe]-hydrogenase genes.

We performed transcriptome sequencing to confirm whether the group B [FeFe]-hydrogenase genes are expressed and likely responsible for the observed activities (Supplementary Dataset 4 and Supplementary Note 1). After 6 hours of fermentative growth, the 7 hydrogenase-encoding *Bacteroides* species had each produced an average of $1.51 \pm 0.6\%$ $H_2$ in their headspace (Supplementary Figs. 1 and 2). The group B [FeFe]-hydrogenase gene was transcribed at high levels during growth of each strain, averaging 180 transcripts per million (TPM) (ranging from $71 \pm 18$ TPM for *Bacteroides fragilis* to $345 \pm 17$ TPM for *Bacteroides plebeius*) (Fig. 3a and Supplementary Dataset 4). Four of

these strains encoded genes for the group B [FeFe]-hydrogenase as their sole $H_2$-metabolizing enzyme. Three other strains (*Bacteroides caccae, Bacteroides thetaiotaomicron* and *Bacteroides faecis*) also encoded genes for the electron-confurcating group A3 [FeFe]-hydrogenase complex, although the transcription of this enzyme was minimal during these growth conditions (average of 3.5 TPM) (Fig. 3a and Supplementary Dataset 4). Importantly, we observed no $H_2$ production by *Bacteroides stercoris*, the only *Bacteroides* species in our isolate collection that consistently lacked detection of any hydrogenase genes (Supplementary Fig. 2). Altogether, these results show that the group B [FeFe]-hydrogenases account for $H_2$ production of *Bacteroides* during fermentative growth and are highly conserved, expressed and active in this genus. Although some *Bacteroides* species have previously been shown to produce $H_2$ (refs. 70–73), our culture-dependent studies indicated that fermentative $H_2$ production is a key feature of their physiology. As elaborated in Supplementary Note 1, patterns of hydrogenase expression and activity varied between species within the class Clostridia; whereas some species appear to primarily dispose of $H_2$ using group B [FeFe]-hydrogenases, fast-growing species (for example, *Clostridium perfringens*) instead appear to be reliant on canonical rapid-acting group A1 [FeFe]-hydrogenases.

We used structural modelling, biochemical measurements and spectroscopy to confirm that group B [FeFe]-hydrogenases are active $H_2$-producing enzymes (Fig. 4). AlphaFold2 modelling (Supplementary Figs. 3 and 4) predicts group B [FeFe]-hydrogenases are structurally conserved among *Bacteroides* species (Supplementary Fig. 5) and distinct from canonical group A1 [FeFe]-hydrogenases (Fig. 4a). As elaborated in Supplementary Note 2, they contain two distinct globular domains: an H-cluster domain (containing the typical catalytic $H_2$-binding H-cluster of [FeFe]-hydrogenases and two [4Fe–4S] clusters) and an unusual smaller ferredoxin-like domain (containing two off-pathway [4Fe–4S] clusters) connected through a short flexible linker (Fig. 4b). We validated that the *Bacteroides* [FeFe]-hydrogenases could bind the catalytic H-cluster and produce $H_2$ by heterologously expressing and semi-synthetically maturing them as previously described (Supplementary Note 3, Supplementary Table 2 and Fig. 7). The group B [FeFe]-hydrogenases expressed from three different species all produced $H_2$ (Fig. 4c), whereas the catalytic subunit of the group A3 enzyme from *B. thetaiotaomicron* showed low activity in this set-up (Fig. 4c and Supplementary Table 2). Despite extensive effort, we were unable to purify stable or active group B [FeFe]-hydrogenases due to the low solubility of these enzymes. This has so far prevented detailed comparisons of their kinetics, electrochemistry or experimental structures compared with group A1 [FeFe]-hydrogenases. Nevertheless, we were able to show through whole-cell X-band electron paramagnetic resonance (EPR) spectroscopy that the *B. fragilis* group B [FeFe]-hydrogenase produced spectroscopic signatures consistent with a typical H-cluster[74,75] (Supplementary Fig. 8, Supplementary Table 3 and Supplementary Note 3). In combination, the structural predictions and recombinant analysis suggest that group B [FeFe]-hydrogenases are true hydrogenases that bind the H-cluster and produce $H_2$, although they differ from other hydrogenases in their redox centres and electron flow pathways.

### *Bacteroides* use group B [FeFe]-hydrogenases to reoxidize ferredoxin during fermentation

We used the transcriptomes of the seven *Bacteroides* species to predict their central carbon metabolism and infer how their [FeFe]-hydrogenases likely integrate into their physiology (Supplementary Dataset 4). Supporting previous observations, these reconstructions suggest that all species are mixed-acid fermenters[76–79] that can break down sugars to pyruvate through the glycolysis pathway, convert pyruvate to acetate (via PFOR and acetate kinase), and also reduce oxaloacetate to succinate (via enzymes including fumarate reductase) and propionate (via the methylmalonyl-CoA pathway).

Consistently, these bacteria all transcribe the genes for these pathways at similarly high levels during mid-exponential fermentative growth on rich media (YCFA) (Fig. 4d). During growth on minimal media (modified *Bacteroides* minimal medium (mBMM)[80]) supplemented with glucose, $H_2$ production and growth were approximately proportional to glucose consumption in *B. fragilis* (Supplementary Fig. 9), indicating that $H_2$ is an important end-product of glucose fermentation through these pathways. $H_2$ was produced concomitantly with millimolar levels of lactate and isobutyrate (Supplementary Fig. 10). Previous metabolic studies have revealed that, in the presence of hemin, *Bacteroides* enhances growth on glucose by producing the cytochrome-containing respiratory fumarate reductases[81–83]. Thus, we examined how fermentation pathways were affected by the availability of hemin. Consistent with previous observations[81–83], metabolism greatly shifted towards more efficient pathways in the presence of hemin, with succinate, propionate and acetate becoming the dominant end-products, in line with the induction of fumarate reductase and the succinate–propionate pathway (Fig. 4d, Supplementary Fig. 10 and Supplementary Dataset 5). $H_2$ production only modestly decreased (by 19%), indicating $H_2$ is still a particularly important sink when respiratory electron acceptors are available. During glucose consumption in the presence of hemin (Fig. 4e), $H_2$ accounted for $9.9 \pm 0.6\%$ of the electrons released during fermentation (Supplementary Dataset 5). This suggests that $H_2$ production is a core part of *Bacteroides* metabolism, aligned with the widespread transcription of the group B [FeFe]-hydrogenase under the hemin-available YCFA media conditions (Fig. 4d).

The group B [FeFe]-hydrogenase potentially reoxidizes the ferredoxin reduced by PFOR during acetyl-CoA production and acetate fermentation, disposing these excess electrons as $H_2$. To confirm this, we performed a biochemical assay using *B. fragilis* cell extracts. The addition of the PFOR substrates CoA and pyruvate to the assay mixture resulted in a 14-fold increase in $H_2$ production ($774 \pm 814$ parts per million (ppm) $H_2$) compared with extracts incubated without added substrates ($0.49 \pm 0.48$ ppm $H_2$) (Fig. 4f), suggesting that increased PFOR activity promotes hydrogenase activity. To test whether this activity was specifically driven by PFOR, control assays were conducted in which cell extracts were pre-incubated with the PFOR inhibitor nitazoxanide (NTZ) before the addition of substrates, resulting in the termination of $H_2$ production. Together, these results suggest that *B. fragilis* PFOR activity supplies the reductant for $H_2$ production and stimulates hydrogenase activity. The Rnf complex, which reversibly couples $Na^+/H^+$ import to reverse electron transport from NADH to oxidized ferredoxin, may provide additional reduced ferredoxin for the hydrogenase[84,85] (Fig. 4d). Other *Bacteroides* fermentation pathways, namely, for succinate, propionate and lactate formation, do not directly compete with $H_2$ production but can divert electrons from PFOR and the pyruvate formate lyase provides an additional PFOR bypass (Fig. 4d). Genes for the succinate/propionate branch and PFOR were relatively consistently transcribed across the strains, whereas pyruvate formate lyase and a putative lactate dehydrogenase expression varied by as much as 4.5-fold and 50-fold, respectively (Supplementary Dataset 4). Thus, lactate may be an important alternative route of electron disposal that competes with $H_2$ production in *Bacteroides* species, in line with the short-chain fatty acid data (Supplementary Fig. 10). The group A3 [FeFe]-hydrogenase may contribute to redox homeostasis by coupling oxidation of reduced ferredoxin and NADH to $H_2$ production under certain conditions (Fig. 4d), but its low transcription suggests a minimal role during fermentative growth.

To further contextualize the importance of the hydrogenases of *Bacteroides*, we interrogated previously published differential fitness datasets that leveraged random barcoded transposon sequencing of *B. thetaiotaomicron* mutant libraries exposed to a range of conditions[86]. In this previous study, transposon mutants were obtained for both hydrogenases, suggesting that they are not essential for growth. However, the group B [FeFe] hydrogenase was important for growth of

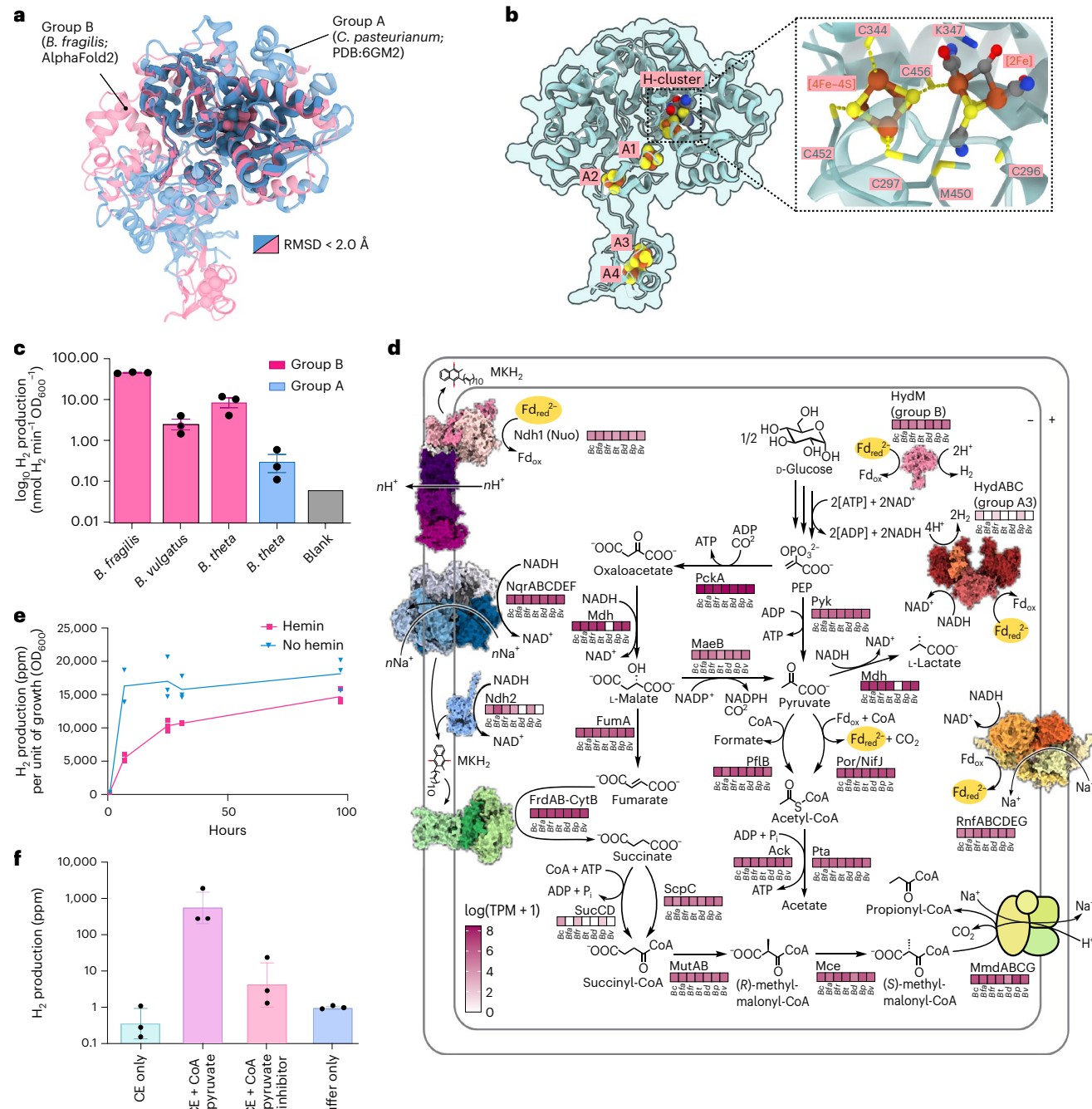

**Fig. 4 | Metabolic integration, predicted structure and biochemical activity of the group B [FeFe]-hydrogenases from *Bacteroides*. a**, Structural superposition of group A [FeFe]-hydrogenase (*Clostridium pasteurianum* CpI; X-ray crystallography; PDB:6GM2) and group B [FeFe]-hydrogenase (*B. fragilis*; AlphaFold2). Structurally similar regions (root mean square deviation (RMSD) < 2 Å) are outlined in black, whereas divergent regions (RMSD > 2 Å) are semi-transparent. **b**, Top-ranked AlphaFold2 model of the *B. fragilis* group B [FeFe]-hydrogenase with modelled putative cofactors. The predicted H-cluster site with coordinating conserved residues is highlighted (green). Four Fe–S clusters (A1–A4) are predicted to coordinate with conserved cysteines throughout the protein. **c**, Average H₂ production (gas chromatography) from lysates (*n* = 3) activated with [2Fe]ᵃᵈᵗ. Results are shown for the group B [FeFe]-hydrogenases of *B. fragilis*, *B. vulgatus* and *B. thetaiotaomicron* (pink), and the group A3 [FeFe]-hydrogenase of *B. thetaiotaomicron* (blue). Activities were normalized for number of cells used (nmol H₂ min⁻¹ OD₆₀₀⁻¹) and error bars reflect s.d. from biological triplicates. All enzymes were expressed in *E. coli* BL21(DE3) cells. 'Blank' represents the same strain but contains an empty vector that was also added with [2Fe]ᵃᵈᵗ (grey). **d**, Average transcription (TPM) of fermentation-

associated genes, including group B and A3 [FeFe]-hydrogenases, across seven enteric *Bacteroides* isolates, in triplicate. **e**, Average *B. fragilis* H₂ production (ppm) relative to growth (OD₆₀₀) with (pink) and without (blue) hemin (*n* = 3). **f**, H₂ production (log₁₀ ppm) of a *B. fragilis* cell extract (CE)-only control (light blue), upon stimulation of the PFOR with CoA, and pyruvate (purple), upon inhibition of PFOR with NTZ (light pink) and a buffer control (dark blue). Error bars indicate s.d. (*n* = 3). Ack, acetate kinase; *Bc*, *B. caccae*; *Bfa*, *B. faecis*; *Bfr*, *B. fragilis*; *Bd*, *B. dorei*; *Bp*, *B. plebeius*; *Bt* and *B. theta*, *B. thetaiotaomicron*; *Bv*, *B. vulgatus*; FrdAB-CytB, fumarate reductase; FumA, fumarase; MaeB, NADP-dependent malic enzyme; Mce, metabolite transporter; Mdh, malate dehydrogenase; MmdABCG, methylmalonyl-CoA decarboxylase; MutAB, methylmalonyl-CoA mutase; Ndh1/Nuo, NADH:ubiquinone oxidoreductase complex I; Ndh2, type II NADH dehydrogenase; NifJ, pyruvate:flavodoxin oxidoreductase; NqrABCDEF, Na⁺-translocating NADH:quinone oxidoreductase; ox, oxidised; PckA, phosphoenolpyruvate carboxykinase; PflB, pyruvate formate-lyase; Por, pyruvate:ferredoxin oxidoreductase; Pta, phosphotransacetylase; Pyk, pyruvate kinase; red, reduced; RnfABCDEG, ferredoxin:NAD⁺ oxidoreductase; ScpC, propionyl-CoA/succinate-CoA transferase; SucCD, succinyl-CoA synthetase.

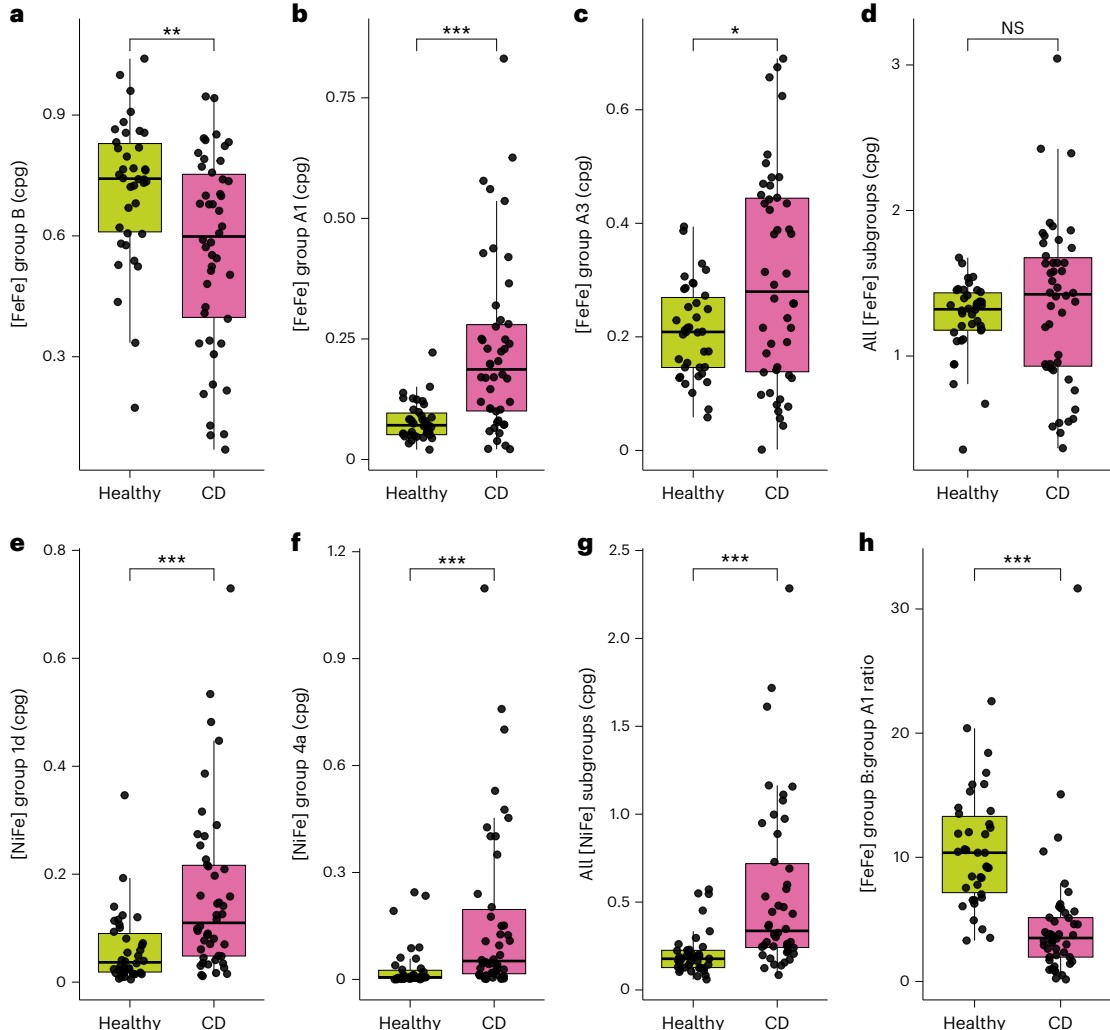

**Fig. 5 | Comparison of hydrogenase gene levels in healthy individuals compared with individuals with Crohn's disease in a case–control study.** **a**–**g**, Sum of counts per genome are shown for genes encoding group B [FeFe]-hydrogenases ($P = 0.0023$) (**a**), group A1 [FeFe]-hydrogenases ($P = 6.6 \times 10^{-7}$) (**b**), group A3 [FeFe]-hydrogenases ($P = 0.04$) (**c**), all [FeFe]-hydrogenase subgroups (non-significant) (**d**), group 1d [NiFe]-hydrogenases ($P = 3.8 \times 10^{-5}$) (**e**), group 4a [NiFe]-hydrogenases ($P = 6.8 \times 10^{-6}$) (**f**), and all [NiFe]-hydrogenase

subgroups (**g**). **h**, Shown is the ratio of group B to group A1 [FeFe]-hydrogenase genes ($P = 1.9 \times 10^{-11}$). Statistical significance was assessed using two-sided Wilcoxon rank-sum tests; *$P < 0.05$, **$P < 0.01$, ***$P < 0.001$. Box plots show the median (centre line), upper and lower quartiles (box limits), 1.5× interquartile range (whiskers) and individual samples. $n = 46$ individuals with Crohn's disease (CD); $n = 38$ healthy controls. NS, not significant.

*B. thetaiotaomicron* on fermentable carbon sources, especially glucose (Supplementary Fig. 11). The hydrogenase was also important for adaptation on specific nitrogen sources such as L-valine (Supplementary Fig. 11) and fitness under stress conditions in rich or defined media. The strongest fitness defect was during treatment with metronidazole, a broad-spectrum prodrug activated by reduced ferredoxin. This is consistent with the predicted role of the group B [FeFe]-hydrogenase in reoxidizing reduced ferredoxin, whereas mutants lacking the enzyme will accumulate in its reduced state and more quickly activate the prodrug. By contrast, transposon mutants interestingly showed increased fitness when exposed to sodium fluoride and fusidic acid. However, the group A3 [FeFe] hydrogenase did not show statistically significant fitness defects under the conditions tested. Altogether, these analyses suggest the group B [FeFe]-hydrogenase enhances the fitness of *Bacteroides* across a range of conditions, although the bacterium can adapt to loss of the enzyme.

## Group B [FeFe]-hydrogenases are depleted in gastrointestinal disorders and other diseases

Finally, to investigate the links between hydrogen metabolism with health and disease, we compared the levels of hydrogenase-associated

genes based on stool metagenomes of 871 healthy individuals and 790 individuals with disease, based on a case–control study of Crohn's disease[87] and reports on 11 other chronic disease phenotypes[55] (Fig. 5). Consistent with the earlier analyses (Fig. 1), group B [FeFe]-hydrogenases were the most abundant hydrogenase genes overall, although their levels often varied between individuals (Fig. 5 and Supplementary Dataset 6). However, their average levels were significantly higher ($P = 0.0023$) in healthy individuals ($0.72 \pm 0.17$ cpg) compared with those with Crohn's disease ($0.56 \pm 0.24$ cpg). Contrastingly, there were strong increases in the average levels of the prototypical fermentative hydrogenase (group A1, 2.8-fold, $P = 6.6 \times 10^{-7}$), formate hydrogenlyase (group 4a, 5.2-fold, $P = 6.8 \times 10^{-6}$), and to a lesser extent the electron-confurcating hydrogenase (group A3, 1.4-fold, $P = 0.04$) in individuals with Crohn's disease. Most notably, the ratio of group B to group A1 [FeFe]-hydrogenase genes shifted by 2.3-fold between the 2 cohorts ($P = 1.9 \times 10^{-11}$). Capacity for $H_2$ oxidation also increased, with a 2.6-fold increase in respiratory group 1d [NiFe]-hydrogenase genes in those with Crohn's disease ($P = 3.8 \times 10^{-5}$) (Fig. 5). Although these differences may be only correlative, altered $H_2$ cycling may contribute to the Crohn's disease phenotype through various possible mechanisms.

For example, intestinal respiratory bacteria (for example, Enterobacteriaceae) may benefit from the elevated $H_2$ production by the highly active group A1 [FeFe]-hydrogenases, by using this electron donor to reduce inflammation-derived electron acceptors. Consistently, a previous study[31] showed increased $H_2$ oxidation contributes to the expansion of *Escherichia coli* during gut inflammation in a murine model. There was also a significant enrichment of group A1 genes compared with group B [FeFe]-hydrogenase genes in several other chronic disease states, including atherosclerosis, liver cirrhosis, colorectal cancer and type 2 diabetes (Supplementary Fig. 12). A range of other significant variations were also observed, including a near absence of group 1d [NiFe]-hydrogenases in type 2 diabetes ($P = 7.8 \times 10^{-9}$) (Supplementary Fig. 13). Further mechanistic studies are required to better understand the basis of these differences.

## Discussion

By integrating analyses at the enzymatic, cellular and gut ecosystem levels, we provide multifaceted evidence that the group B [FeFe]-hydrogenase mediates $H_2$ production in diverse bacteria and drives fermentation in the healthy human gut. These observations also suggest that *Bacteroides*, a genus previously unrecognized as a major $H_2$ producer, plays a more central role in gut $H_2$ cycling than initially understood and uses a hydrogenase of previously unknown function. It remains unclear what competitive advantage is conferred by the group B [FeFe]-hydrogenase compared with the functionally similar group A1 [FeFe]-hydrogenase. Both enzymes are predicted to be monomeric ferredoxin-dependent $H_2$-producing enzymes with similar active site structures and biosynthetic pathways. Nevertheless, the group B enzyme is unique for its ferredoxin-like domain separated by a flexible linker and also seems to have somewhat lower activity than its group A1 counterparts based on the cellular and whole-cell data. Detailed side-by-side studies of the protein–protein interactions, kinetics, electrochemistry and oxygen sensitivity of the purified enzymes may help disentangle their differences. Nevertheless, it is apparent that the group B enzyme has been selected in diverse bacterial species to produce high levels of $H_2$ during fermentative growth[44]. Genes encoding this hydrogenase are particularly abundant in healthy people and may be an indicator of $H_2$ homeostasis, whereas there is a shift in favour of group A1 [FeFe]-hydrogenases in disease states such as Crohn's disease.

This study also provides a holistic perspective on the microorganisms and enzymes responsible for $H_2$ cycling in the human gut. We provide an in-depth study of the distribution of gastrointestinal hydrogenases, surpassing our last bioinformatics survey in this area that was limited to just 20 metagenomes[19], and bridge genomic insights with culture- and enzyme-based validation. We show $H_2$ production is an extremely widespread trait, extending to gut Bacteroidetes, Fusobacteria, Actinobacteria and Proteobacteria in addition to the well-studied Clostridia of the phylum Firmicutes. Our findings also highlight that the mediators of the three conventionally described pathways for $H_2$ disposal, namely, methanogenesis, acetogenesis and sulfidogenesis, are in low abundance but are transcriptionally active in the gut[3,12,13]. Instead, it is nevertheless probable that respiratory bacteria that use electron acceptors such as fumarate, nitrate and sulfoxides play a potentially dominant role in gut $H_2$ consumption, given the abundance and transcription of associated hydrogenases and respiratory reductases in both stool and biopsy metagenomes and stool metatranscriptomes. However, activity measurements are required to validate this. In addition, recent work[88] has highlighted the potential for numerous additional organic electron acceptors in the gut, including dietary and host-derived compounds, that could also be involved in $H_2$ oxidation. Enterobacteriaceae may be particularly important hydrogenotrophs. These lineages are especially enriched in certain disease states, such as Crohn's disease, where they may support both aerobic and anaerobic respiration using inflammation-derived electron acceptors. Follow-up studies should combine metagenomic, biochemical and culture-based studies to determine which processes and microorganisms dominate $H_2$ oxidation in the human gut. Further studies are also required to better characterize the roles of some of the moderately abundant but still functionally characterized hydrogenases identified here, including the group A2 [FeFe]-hydrogenases, group 4e [NiFe]-hydrogenases and the sensory hydrogenases. There is also a critical need to better understand what drives the vast interindividual variation in hydrogenase composition and expression between individuals, and how this relates with gastrointestinal function and disease states. In summary, our multifaceted approach uncovers the abundance, diversity and functional roles of hydrogenases of previously unrecognized importance in the human gut during health and disease.

## Methods

### Mucosal biopsy sampling and metagenomes

Mucosal biopsy metagenome samples were obtained from 102 mucosal biopsy enrichment metagenomes of 42 paediatric patients receiving colonoscopies due to non-inflammatory conditions at the Monash Children's Hospital, after receiving written consent (Monash Health; Human Research Ethics Committee (HREC/16/MonH/253) and Monash University Ethics Committee (Monash Health reference 16367A)). Samples were obtained from the terminal ileum, caecum and rectum and transferred to anaerobic conditions within 15 min of collections. Biopsy metascrapes[68] were performed using biopsy samples, serially diluted to $10^{-6}$ using sterile pre-reduced 1× PBS, plated directly onto YCFA agar and evenly spread using a sterile disposable plate spreader (International Scientific Group). Plates were incubated at 37 °C for 24 h under anaerobic conditions. Plates showing distinct, well-separated bacterial colonies were selected for scraping, whereby 600 µl of pre-reduced 1× PBS was added to the plate surface and bacterial colonies were resuspended using a sterile spreading loop (International Scientific Group). The resulting bacterial suspension was collected into microcentrifuge tubes (Eppendorf) and stored at −80 °C until DNA extraction using the MP Biomedicals FastDNA SPIN Kit for soil and sequenced on the Illumina NextSeq2000. Resulting data are accessible via the European Nucleotide Archive (ENA) under accession number PRJEB45397.

### Metagenomic and metatranscriptomic analyses

For the stool metagenome and metatranscriptome analyses, raw paired-end short reads were obtained from an inflammatory bowel disease microbiota functionality study[57], accessed via the ENA under accession number PRJNA389280. The dataset included 78 paired metagenomes and metatranscriptomes, along with 222 additional metagenomes from the faecal samples of 117 patients. The stool metagenome, stool metatranscriptome and mucosal biopsy metagenomes were quality checked using FastQC (v.0.11.7) (https://www.bioinformatics.babraham.ac.uk/projects/fastqc/)[89] and MultiQC (v.1.0)[90]. Adaptor and PhiX sequences and low-quality bases were trimmed and filtered using BBDuk from the BBTools (v.38.51) suite[91]. SortmeRNA (v.4.3.3)[92] removed rRNA sequences from metatranscriptomic reads. Resulting cleaned forward reads were screened (blastx) using DIAMOND v.2.0 (ref. 93) against the HydDB dataset[61] and a manually curated in-house database including enzymes associated with $H_2$-producing and $H_2$-consuming pathways[94]. Alignments were filtered to a minimum length of 28 amino acids, and then further filtered based on the following minimum percentage identity thresholds previously determined and validated for each protein in the database: 50% for AcsB, ArsC, AsrA, CcoN, CooS, CoxA, CydA, CyoA, DsrA, FdhA, NapA, NarG, NiFe (60% for group 4), NifH, NirK, NorB, NosZ, NrfA, RHO, SdhA/FrdA and Sqr; 60% for FeFe and NuoF; and 70% for AtpA and YgfK. Read counts were normalized to RPKM, and metagenomes were further normalized against mean RPKM values estimated from 14 single-copy ribosomal marker genes to obtain an 'average gene copy per organism' value for each gene. For predicted hydrogenase sequence reads, the taxonomy of the best hit was retrieved and summarized in RPKM

to evaluate which taxonomic groups contribute most of these reads. To understand the relative abundance of microbial taxa detected throughout the stool and biopsy metagenomic samples, the paired-end sequencing reads were first merged and then taxonomic profiling was performed using MetaPhlAn4 (v.4.0.6)[95]. The resulting MetaPhlAn4 output files were processed using the accompanying Python script (merge_metaphlan_tables.py) to generate relative abundance tables for downstream analysis of the prevalence and average abundance of *Bacteroides*.

### Gut isolate genomic analysis

Whole-genome sequences of 818 gut isolates from adult and paediatric faecal and biopsy samples were obtained from the Australian Microbiome Culture Collection (AusMiCC; https://ausmicc.org.au/) and a previous study[96] describing a collection of gut isolate genomes (Human Gastrointestinal Bacteria Genome Collection). Genomes are also accessible via the ENA under project accessions PRJEB70412 and PRJEB23845 (with run accession numbers listed in Supplementary Dataset 2). Genomes were quality checked using CheckM (v.1.1.3)[97] and those with >90% completeness and <5% contamination ($n = 812$) were retained. Protein sequences of the retained genomes were used to search for and identify alignments matching the previously mentioned protein database using the blastp function of DIAMOND (v.2.0.9)[93]. Alignment criteria included query and subject coverage thresholds set at 80%, and further filtering was conducted based on the previously mentioned percentage identity thresholds for each database protein. GTDB-Tk (v.1.6.0; database R06-RS202)[98] was used to assign a taxonomic classification to each isolate using the 'classify_wf' option, and a phylogenetic tree was constructed with the 'de_novo_wf' option. The tree was visualized, midpoint-rooted, and the copy number per isolate genome of relevant hydrogen-related metabolic genes and hydrogenase subgroups was overlaid using the Interactive Tree of Life (iTOL)[99] to observe differences in hydrogen metabolism across the different phylogenetic groups.

### Bacterial growth analyses

All isolates used in this study, sourced from healthy human faecal samples, were obtained from AusMiCC[68,96]. Nineteen isolates were selected to compare the expression and activity of the group B [FeFe]-hydrogenases with other $H_2$-producing bacteria across taxonomically diverse gut bacteria, including three control isolates lacking hydrogenases and three positive controls encoding well-characterized $H_2$-producing hydrogenases[42,100]. Supplementary Table 1 lists the strains and their hydrogenase content. All isolates were accessed from glycerol stocks containing YCFA broth media[101] with 25% glycerol, stored at −80 °C, and revived in pre-reduced YCFA broth. Incubation was carried out anaerobically at 37 °C in an atmosphere of 10% $H_2$, 10% $CO_2$, 80% $N_2$ for 24 h. Solid media, when required, were supplemented with 0.8% w/v of bacterial agar. Growth assessment involved measuring the optical density at a wavelength of 600 nm ($OD_{600}$) of each isolate over 24 h while anaerobically cultured in YCFA broth at 37 °C. Each isolate, in duplicate, was subcultured into a 200 μl 96-well plate, with a 1:100 dilution of culture to broth. An hourly assessment of $OD_{600}$ was conducted using a FLUOstar Omega Microplate Reader, with readings taken under anaerobic conditions, and shaking before each measurement.

### Hydrogen production assays

For the $H_2$ production assay, isolates were plated on pre-reduced YCFA agar plates and grown anaerobically at 37 °C for 24 h. Single colonies were used to inoculate 3 ml of pre-reduced YCFA broth in a 15 ml Falcon tubes, which were incubated anaerobically at 37 °C for 24 h. After incubation, each separate starter culture was used to inoculate triplicate 30 ml aliquots of YCFA broth to a starting $OD_{600}$ of 0.025. Cultures were maintained in 120 ml glass serum vials sealed with laboratory-grade butyl rubber stoppers. Immediately after inoculation, the headspace of

each culture vial was flushed for 10 min with 99.99% pure $N_2$ to remove residual $H_2$ and ensure that production of $H_2$ was thermodynamically favourable and entirely biotic in origin. Gas chromatography was used to assess the $H_2$ production capabilities of each isolate over time. To establish a baseline $H_2$ concentration for each isolate (in triplicate), a gas-tight syringe was used to collect initial headspace gas samples from each culture immediately after $N_2$ flushing. Headspace gas samples were then collected at predetermined time points based on growth curve data and until increases in $H_2$ concentration were no longer detected. $H_2$ concentration was measured using a gas chromatograph containing a pulsed discharge helium ionization detector (model TGA-6791-W-4U-2; Valco Instruments Company) that uses injection, backflushing, sample selection valves and column-associated sample separation (HayeSep Db at 55 °C, following pre-column separation using a 5 Å molecular sieve at 140 °C)[102]. $H_2$ concentrations were regularly calibrated against ultra-pure $H_2$ standards. This gas chromatograph was able to detect a wide range of $H_2$ concentrations (0.1–10% $H_2$), although sample dilution of 2.5× was necessary to measure the $H_2$ produced by the isolates within the quantifiable range. Calibration samples of known $H_2$ concentration were used to quantify $H_2$ in ppm. The $H_2$ concentration within the media-only control vials was measured concurrently to confirm that $H_2$ production in isolate samples was biotic.

### Glucose consumption assays

*B. fragilis* isolate CC01400 (AusMiCC) was accessed from glycerol stocks containing YCFA broth media with 25% glycerol, stored at −80 °C and revived on a pre-reduced YCFA plate. Single colonies were used to inoculate 3 ml of pre-reduced YCFA broth in 15 ml Falcon tubes, which were incubated anaerobically at 37 °C for 24 h. After incubation, each separate starter culture was used to inoculate triplicate 30 ml aliquots of mBMM[80] broth to a starting $OD_{600}$ of 0.025. Cultures were maintained in 120 ml glass serum vials sealed with laboratory-grade butyl rubber stoppers. Immediately after inoculation, the headspace of each culture vial was flushed for 10 min with 99.99% pure $N_2$. When solid media were required, mBMM was supplemented with 2.5 mg ml$^{-1}$ glucose and 0.8% w/v agar. To evaluate the effect of glucose concentration on growth and $H_2$ production, mBMM supplemented with 0 mg ml$^{-1}$, 2.5 mg ml$^{-1}$ or 5 mg ml$^{-1}$ glucose was used. Growth was monitored by measuring $OD_{600}$ over 30 h, and 2 ml headspace gas samples were taken in parallel using gas-tight syringes and stored in 3 ml Exetainers (2.5× dilution). Glucose levels in culture supernatant were quantified using the Invitrogen Glucose Colorimetric Detection Kit (catalogue number EIAGLUC). To remain within the assay's detection range (0.5–32 mg 100 ml$^{-1}$), samples from the 5 mg ml$^{-1}$ glucose condition at ≤6 h were diluted 25-fold, and those at 24 h were diluted 12.5-fold. Samples from the 2.5 mg ml$^{-1}$ condition at ≤6 h were also diluted 12.5-fold. Absorbance at 560 nm was measured using a FLUOstar Omega Microplate Reader. Gas samples were measured using a gas chromatograph containing a pulsed discharge helium ionization detector (model TGA-6791-W-4U-2; Valco Instruments Company). To evaluate the effect of hemin addition, mBMM broth was supplemented with 2.5 mg ml$^{-1}$ glucose, with or without the addition of 5 μg ml$^{-1}$ hemin. Growth and $H_2$ production were assessed over time by taking samples of culture and headspace gas and monitoring $OD_{600}$ spectroscopically and $H_2$ concentrations via the gas chromatograph. In addition, glucose consumption was also quantified as mentioned earlier.

### Volatile fatty acid analysis

Triplicate cultures were collected during stationary phase (with-hemin condition, average $OD_{600}$ of 0.3; without-hemin condition, average $OD_{600}$ of 0.5). Culture (30 ml) was pelleted via centrifugation (4,500$g$, 15 min at 4 °C) and the resulting supernatant was filtered with a 0.2 μm Acrodisc syringe filter (Pall Life Sciences) and collected into 4 ml furnaced borosilicate glass vials with Teflon-lined lids (Sigma-Aldrich). Samples were stored at −20 °C and thawed before

analysis. Recrystallized 2-nitrophenyl hydrazine (NPH; Sigma-Aldrich) was used to prepare a 0.1 M NPH solution by dissolving in 0.25 M HCl. N-(3-dimethylaminopropyl)-N′-ethylcarbodiimide hydrochloride (EDC; Sigma-Aldrich) was used to prepare a 0.3 M solution by dissolving in Milli-Q water. Equal volumes of redistilled pyridine (Sigma-Aldrich) and concentrated HCl were mixed to prepare a pyridine buffer. Standards and samples were treated as follows to obtain the NPH derivatives of the volatile fatty acids: to each 2 ml sample, 200 µl of pyridine buffer was added and bubbled for 4 min to remove any $CO_2$. Then 200 µl each of NPH and EDC was added, mixed and incubated for 1.5 h at room temperature. Following that, 200 µl of 40% (w/v) KOH was added to each sample, and samples were mixed and incubated at 70 °C for 10 min (ref. 103). Samples were left to cool down and settle overnight before analysis. Treated standards and samples were analysed via reversed-phase high-performance liquid chromatography using an Agilent SB-C8 preconcentrator and guard column (4.6 mm × 12.5 mm) and analytical column (4.6 mm × 250 mm). Volatile fatty acid concentrations were determined by comparing retention times and peak areas with treated standards, and then normalizing the concentration to growth for each replicate.

## RNA extraction

Transcriptomic analysis was performed for all isolates (except *B. stercoris*) to verify the active transcription of hydrogenase genes identified within the genome. Isolates were accessed from glycerol stocks containing YCFA broth media with 25% glycerol, stored at −80 °C, and revived on a pre-reduced YCFA plate. Single colonies were used to inoculate 3 ml of pre-reduced YCFA broth in 15 ml Falcon tubes, which were incubated anaerobically at 37 °C for 24 h. After incubation, each separate starter culture was used to inoculate triplicate 30 ml aliquots of YCFA broth to a starting $OD_{600}$ of 0.025. Cultures were maintained in 120 ml glass serum vials sealed with laboratory-grade butyl rubber stoppers. Immediately after inoculation, the headspace of each culture vial was flushed for 10 min with 99.99% pure $N_2$. Cells were collected for RNA extraction during active $H_2$ production at either exponential phase (isolates with a group A or B [FeFe]-hydrogenase) or stationary phase (isolates with a group 4a [NiFe]-hydrogenase), as indicated by previously conducted growth curves. To quench cells, a glycerol–saline solution (3:2 v/v, −20 °C) was added before centrifugation (4,500g, 30 min at −9 °C). The cell pellet was resuspended in 1 ml of an additional glycerol–saline solution (1:1 v/v at −20 °C) and centrifuged again (4,500g, 30 min at −9 °C). Cell pellets were then resuspended in 1 ml TRIzol reagent, transferred to a tube containing 0.3 g of 0.1 mm zircon beads, and subjected to 5 cycles of bead-beating (30 s per cycle, 5,000 rpm, resting on ice for 30 s between cycles) using a Bertin Technologies Precellys 24 bead-beater before centrifugation (12,000g, 10 min at 4 °C). The supernatant was transferred to a new tube and 200 µl of chloroform was added, inverted to mix for 15 s, and then incubated at room temperature for 2–3 min before centrifugation (10,000g, 15 min at 4 °C) for phase separation. The aqueous phase underwent purification using the RNeasy Mini Kit following the manufacturer's instructions (Qiagen), with on-column DNA digestion using the RNase-free DNase Kit (RNeasy Mini Handbook; Qiagen). RNA was eluted into RNase-free water, and the concentration for each sample was determined using the RNA HS Qubit Assay Kit according to manufacturer's instruction (Thermo Fisher Scientific).

## Transcriptome sequencing

The Monash Health Translation Precinct Medical Genomics Facility prepared libraries using the Illumina Stranded Total RNA prep with Ribo-Zero plus Microbiome kit. A total of 200 ng of RNA underwent 16 cycles of amplification. Final libraries were quantified by Qubit, combined into an equimolar pool, and quality checked by Qubit, Bioanalyzer and quantitative PCR. For sequencing, 1,000 pM of the library pool was clustered on a P2 NextSeq2000 run and 59 bp sequencing

was performed. The total run yield was 66.56 Gb, with approximately 496.7 million reads passing the filter, achieving 92.57% of bases with a Phred quality score of at least 30 (%Q30). Transcriptomic data were quality checked and pre-processed using FastQC (v.0.11.7)[89], MultiQC (v.1.0)[90] and BBDuk from the BBTools suite (v.38.51)[91]. Successful ribodepletion was confirmed by SortMeRNA (v.4.3.3)[92]. Each isolate's genome was annotated using Prokka (v.1.14.6)[104], and transcription was quantified by mapping the transcripts to these annotated genomic features using Salmon (v.1.9.0)[105] with default settings (salmon quant). Gene transcription was quantified as relative abundance in TPM. To identify transcripts matching previously identified hydrogenase hits, Prokka-generated annotated protein sequence files were validated with DIAMOND alignment as described earlier. Transcript identifiers were used to match the hydrogenase hits for each isolate to the corresponding TPM values for evaluation of hydrogenase transcription. For metabolic pathway analysis, DRAM (v.1.4.6)[106] was used to annotate each transcriptome with the KEGG protein database[107]. Transcripts are available via the ENA under project accession number PRJEB70412, with run accession numbers listed in Supplementary Dataset 4.

## PFOR and hydrogenase assay in cell extracts

All steps were performed under anaerobic conditions (100% $N_2$) unless otherwise specified. To determine if the group B [FeFe]-hydrogenase activity is coupled to the PFOR, *B. fragilis* cells (50 ml culture, $n = 3$, late exponential phase) were collected by centrifugation at 4,500g for 30 min at 4 °C. The cell pellets were resuspended in 50 mM Tris, pH 7.0, and 200 mM NaCl, re-pelleted by centrifugation, frozen in liquid $N_2$ and stored at −80 °C until further use. The cell pellets were thawed and resuspended in 10 ml 50 mM HEPES pH 7.5, with 2 mM $MgCl_2$, 2 mM dithiothreitol (DTT) and 0.1 mg ml⁻¹ DNase. The cells were lysed using a Constant Systems cell disruptor (2× at 40,000 psi) and the lysate was centrifuged at 12,000g for 45 min at 4 °C. The supernatant was passed through a 0.2 µm filter and the protein concentration was estimated using the Bradford assay, yielding 4.73 mg ml⁻¹ protein in a total of 11 ml lysate. The activity assays were performed twice in triplicates under an $N_2$ atmosphere in 10 ml serum vials, sealed with butyl rubber stoppers and aluminium crimps. The assay mixture contained 10 ml anaerobic 50 mM HEPES pH 7.5, with 2 mM $MgCl_2$ and 2 mM DTT. The reaction was started by the addition of 0.131 mg cell extract, followed by incubation at 37 °C for 24 h in the dark. To stimulate PFOR activity and generation of reduced ferredoxin, sodium pyruvate (10 mM final concentration) and CoA sodium salt hydrate (0.2 mM final concentration) were added before the addition of cell extract. To test whether the observed $H_2$ production upon pyruvate and CoA addition was dependent on PFOR activity, reactions were also performed in the presence of the PFOR inhibitor NTZ. For inhibition, 1 ml of cell extract was pre-incubated with 100 µM NTZ for 5 min before addition to the reaction mixture containing pyruvate and CoA. Hydrogenase activity was assessed by quantifying $H_2$ production in the vial headspace using gas chromatography (model TGA-6791-W-4U-2; Valco Instruments Company) equipped with a pulsed discharge helium ionization detector.

## AlphaFold2 structural modelling

Protein structure predictions from *Bacteroides* group B [FeFe]-hydrogenase sequences were generated using AlphaFold2 (v.2.1.1)[108,109] through the ColabFold (v.1.5.2)[110] notebook. The specified ColabFold parameters were as follows: num_relax (1), template_mode (none), msa_mode (mmseqs2_uniref_env), pair_mode (unpaired_paired), model_type (alphafold2_ptm) and pairing_strategy (greedy). For the *B. fragilis* group B [FeFe]-hydrogenase model (*Bf*HydM), num_recycles was set to 48, whereas *B. thetaiotaomicron* and *Bacteroides vulgatus* num_recycles were set to 3. For the *B. thetaiotaomicron* group A3 [FeFe]-hydrogenase model (*Bt*HydABC), num_recycles was set to 48. To model cofactors into the predicted *Bf*HydM and *Bt*HydABC apo structures, the Foldseek[111] web server was used to search the PDB100

database for experimental structures with similar folds to *Bf*HydM and *Bt*HydABC. The following Foldseek parameters were used: databases (PDB100 2201222), mode (3Di/AA) and taxonomic filter (none). For *Bf*HydM, two experimental structures returned by Foldseek, PDB 8ALN[112] and 1FCA[113], showed high structural similarity to the input, while also containing iron–sulfur clusters and an H-cluster (Supplementary Fig. 4). Similarly, for *Bt*HydABC, three experimental structures returned by Foldseek were used for cofactor modelling, PDB 8A5E[64], 1FEH[40] and 1FCA[113] (Supplementary Fig. 6). UCSF ChimeraX (v.1.6.1)[114] was used to align these experimental structures to the predicted *Bf*HydM and *Bt*HydABC models with the matchmaker command (Needleman–Wunsch algorithm setting). Cofactors were added in positions corresponding to those of the experimental structures, as shown in Supplementary Figs. 3 and 6. At sites where the AlphaFold2 model and the experimental structures differed, cofactors were manually positioned and adjusted to optimize coordination and to minimize clashes, and bond lengths were assessed to ensure they were biochemically valid.

### Protein expression and preparation

Chemicals used for protein production and characterization were purchased from VWR and used as received unless otherwise stated. Genes encoding the group B [FeFe]-hydrogenases of *B. fragilis*, *B. vulgatus*, and *B. thetaiotaomicron* and the group A3 [FeFe]-hydrogenase of *B. thetaiotaomicron* (Supplementary Table 2) were cloned into pET-11a(+) by Genscript, using restriction sites NdeI and BamHI following codon optimization for expression in *E. coli*. Chemically competent *E. coli* BL21(DE3) cells were transformed using the constructs to express the apo forms of the hydrogenases lacking the di-iron subsite of the H-cluster. Triplicate starter cultures were grown overnight in 5 ml LB medium containing 100 µg ml$^{-1}$ ampicillin at 37 °C. These cultures were subsequently used to inoculate triplicate 80 ml aliquots of M9 medium (22 mM Na$_2$HPO$_4$, 22 mM KH$_2$PO$_4$, 85 mM NaCl, 18 mM NH$_4$Cl, 0.2 mM MgSO$_4$, 0.1 mM CaCl$_2$, 0.4% (v/v) glucose) containing 100 µg ml$^{-1}$ ampicillin. Cultures were grown at 37 °C and 150 rpm until reaching an OD$_{600}$ of approximately 0.4 to 0.6. Protein expression was induced by the addition of 0.1 mM FeSO$_4$ and 1 mM IPTG. Induced cultures were incubated at 20 °C and 150 rpm for approximately 16 h. Cells were thereafter collected by centrifugation at 4,930$g$ for 10 mins at 4 °C. All subsequent operations were carried out under anaerobic conditions to prevent hydrogenase inactivation by atmospheric oxygen in an MBRAUN glovebox (O$_2$ concentration < 5 ppm). The cell pellets were resuspended in a 0.5 ml lysis buffer (30 mM Tris-HCl pH 8.0, 0.2% (v/v) Triton X-100, 0.6 mg ml$^{-1}$ lysozyme, 0.1 mg ml$^{-1}$ DNase, 0.1 mg ml$^{-1}$ RNase). Cell lysis involved 3 cycles of freezing and thawing in liquid N$_2$, and the supernatant was recovered by centrifugation (29,080$g$, 10 mins at 4 °C).

### H$_2$ production assays of activated hydrogenases

The H$_2$ production assays followed established protocols with minor modifications[115]. In short, the [2Fe]$_H$ subsite mimic, (Et$_4$N)$_2$[Fe$_2$(µ-SCH$_2$NHCH$_2$S)(CO)$_4$(CN)$_2$] ([2Fe]$^{adt}$), was synthesized in accordance with previous protocols with minor modifications and verified by Fourier transform infrared spectroscopy[116,117]. Incorporation of cofactor involved the addition of 100 µg of the [2Fe]$^{adt}$ subsite mimic (final concentration 80 µM) to 380 µl of the supernatant in potassium phosphate buffer (100 mM, pH 6.8) and 1% (v/v) Triton X-100. The reaction mixture was anaerobically incubated at 20 °C for 1–4 h in a sealed vial. The non-purified lysate containing the [2Fe]$^{adt}$ subsite mimic was mixed with 200 µl of potassium phosphate buffer (100 mM, pH 6.8) with 10 mM methyl viologen and 20 mM sodium dithionite. Reactions were incubated at 37 °C for up to 120 min. H$_2$ production was determined by analysing the reaction headspace after 15 min using a PerkinElmer Clarus 500 gas chromatograph equipped with a thermal conductivity detector and a stainless-steel column packed with molecular sieve (60/80 mesh). The operational temperatures of

the injection port, oven and detector were 100 °C, 80 °C and 100 °C, respectively. Argon was used as carrier gas at a flow rate of 35 ml min$^{-1}$. The strain expressing the prototypical group A1 [FeFe]-hydrogenase (*Chlamydomonas reinhardtii*, *Cr*HydA1)[115,118–121] served as a positive control, while 'blank' denoted the same strain but one containing an empty vector that was also added with [2Fe]$^{adt}$. Three biological replicates were run at varying times (1–4 h) of incubating the cell lysates with the [2Fe]$^{adt}$ subsite mimic. Incubation time was not found to influence the observed H$_2$ production. Thus, variation in H-cluster formation rates did not appear to have a substantial influence on the outcome of the screening process.

### Whole-cell EPR spectroscopy

Samples for whole-cell EPR spectroscopy were prepared following a previously published protocol with minor modifications[115]. The cell pellet from 80 ml cultures (see 'Protein expression and preparation') was resuspended in 1 ml M9 medium, flushed with N$_2$ gas for 10 min and mixed with a [2Fe]$_H$ subsite mimic that lacks the natural nitrogen bridgehead of [2Fe]$^{adt}$ to propane-1,3-dithiolate ([2Fe]$^{pdt}$, (Et$_4$N)$_2$[Fe$_2$(µ-SCH$_2$CHCH$_2$S)(CO)$_4$(CN)$_2$]). This alternative mimic was synthesized according to previous protocols with minor modifications and verified by Fourier transform infrared spectroscopy[119–121]. The dense cell suspension was centrifuged and the cell pellet was washed with 1 ml Tris-HCl buffer (100 mM Tris, 150 mM NaCl, pH 8.0) 3× under anaerobic conditions. The cells were then resuspended with 200 µl Tris buffer, pH 8.0, and transferred into EPR tubes. The tubes were capped and promptly frozen in liquid N$_2$. Measurements were performed on a Bruker ELEXYS E500 spectrometer using an ER049X SuperX microwave bridge in a Bruker SHQ0601 cavity equipped with an Oxford Instruments continuous flow cryostat and using an ITC 503 temperature controller (Oxford Instruments).

### Metagenomic analyses across health status

To assess the distribution of hydrogenases across health status, we used a previously curated and quality controlled dataset containing 1,661 metagenomes from 33 studies[55]. The dataset encompassed 871 healthy and 790 individuals with disease, including 11 chronic disease phenotypes. Quality control was performed with TrimGalore v.0.6.6 (ref. 122) using a threshold of 80 bp for read length and minimum Phred score of 25. Host sequence reads were removed by mapping the sequence reads to the human genome with bowtie v.2.3.552 (ref. 123). To minimize the impact of sequence depth, samples were rarefied to 15M reads with seqtk v.1.3 (ref. 124), as previously described[55]. Forward reads were mapped to the HydDB dataset[61] and a manually curated in-house database including enzymes associated with H$_2$-producing and H$_2$-consuming pathways[94] with DIAMOND v.2.0 (ref. 93). Alignments were filtered to a minimum length of 26 amino acids, subject to identity threshold filtering, normalized to RPKM, and then again against mean RPKM values estimated from 14 single-copy ribosomal marker genes to obtain an average gene copy per organism value for each gene. The largest case–control inflammatory bowel disease-related study within this dataset[87] was selected to investigate the distribution of hydrogenase subgroups between healthy and disease-associated microbiota, which included 46 patients with Crohn's disease and 38 healthy controls. Statistical significance was assessed with two-sided Wilcoxon rank-sum tests, using the Holm–Bonferroni method to account for multiple comparisons across disease states.

### Reporting summary

Further information on research design is available in the Nature Portfolio Reporting Summary linked to this article.

## Data availability

The new metagenomes, genomes and transcriptomes analysed in this study have been uploaded to ENA under accession numbers

PRJEB45397 (biopsy metagenomes) and PRJEB70412 (isolate genomes and transcriptomes). Source data are provided in Excel tables for each figure.

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

## Acknowledgements

This study was supported by an NHMRC EL2 Fellowship (APP1178715; to C.G.), an ARC Future Fellowship (FT240100502;

to C.G.), the Victorian Government's Operational Infrastructure Support Program, an Australian Government Research Training Scholarship (to C.W.), the Swedish Energy Agency (48574-1 to G.B.), the Swedish Research Council (to G.B.) and the Novo Nordisk Foundation (NNF21OC0066716 to H.L.). R.L. and V.R.M. are supported by ARC DECRA Fellowships (DE230100542 and DE220100965). D.L. is supported by Australian Research Council Laureate Fellowship (FL210100258). S.C.F. is supported by a CSL Centenary Fellowship.

## Author contributions

C.G., C.W., S.C.F., D.L., P.G.W., H.R.G. and J.M.R. conceptualized this study. C.G., S.C.F., D.L., C.W., R.L., T.D.W., H.L. and G.B. supervised students. C.G., C.W., S.C.F., G.B., H.L., R.L., T.D.W., I.C., E.C.M. and M.J. designed experiments. C.W. conducted most experiments and analysed most data. Specific authors contributed to metagenomic screening (C.W., C.G. and R.L.), genomic screening (C.W., S.C.F. and E.L.G.), biopsy metagenome collection (G.L.D., E.M.G., S.C.F. and R.B.Y.), culture-based gas assays (C.W., M.J., N.Q.D., N.B., J.S., E.L.G., S.C.F. and C.G.), transcriptomics (C.W., D.J.K., J.A.G., S.C.F., T.D.W., G.V.P. and C.G.), biochemical characterization (P.R.C., G.B., P.H., H.L., K.W. and C.G.), structural modelling (J.P.L., C.G. and D.J.K.), short-chain fatty acid analysis (T.K.) and analyses of disease states (V.R.M., C.G., C.W., S.C.F. and R.B.Y.). C.W., C.G., T.D.W. and R.L. wrote the paper with input from all authors.

## Funding

## Competing interests

S.C.F. is a co-founder and R.B.Y. is an employee of BiomeBank. The other authors declare no competing interests.

## Additional information

**Correspondence and requests for materials** should be addressed to Caitlin Welsh, Samuel C. Forster or Chris Greening.

[1]Department of Microbiology, Biomedicine Discovery Institute, Monash University, Clayton, Victoria, Australia. [2]Centre for Innate Immunity and Infectious Disease, Hudson Institute of Medical Research, Clayton, Victoria, Australia. [3]Department of Molecular and Translational Sciences, Monash University, Clayton, Victoria, Australia. [4]Department of Chemistry, Ångström Laboratory, Uppsala University, Uppsala, Sweden. [5]Melbourne Integrative Genomics, School of BioSciences, University of Melbourne, Parkville, Victoria, Australia. [6]Department of Microbiology and Immunology at the Peter Doherty Institute for Infection and Immunity, University of Melbourne, Parkville, Victoria, Australia. [7]Department of Chemistry and Chemical Biology, Harvard University, Cambridge, MA, USA. [8]Carl R. Woese Institute for Genomic Biology, University of Illinois Urbana-Champaign, Urbana, IL, USA. [9]Department of Microbiology and Immunology, University of Michigan Medical School, Ann Arbor, MI, USA. [10]Department of Nutrition Science, College of Health and Human Sciences, Purdue University, West Lafayette, IN, USA. [11]Department of Paediatrics, School of Clinical Sciences at Monash Health, Monash University, Clayton, Victoria, Australia. ✉e-mail: cait.welsh@monash.edu; sam.forster@hudson.org.au; chris.greening@monash.edu

# Reporting Summary

## Statistics

For all statistical analyses, confirm that the following items are present in the figure legend, table legend, main text, or Methods section.

| n/a | Confirmed | |
|---|---|---|
| ☐ | ☒ | The exact sample size (*n*) for each experimental group/condition, given as a discrete number and unit of measurement |
| ☐ | ☒ | A statement on whether measurements were taken from distinct samples or whether the same sample was measured repeatedly |
| ☐ | ☒ | The statistical test(s) used AND whether they are one- or two-sided<br>*Only common tests should be described solely by name; describe more complex techniques in the Methods section.* |
| ☐ | ☒ | A description of all covariates tested |
| ☐ | ☒ | A description of any assumptions or corrections, such as tests of normality and adjustment for multiple comparisons |
| ☐ | ☒ | A full description of the statistical parameters including central tendency (e.g. means) or other basic estimates (e.g. regression coefficient) AND variation (e.g. standard deviation) or associated estimates of uncertainty (e.g. confidence intervals) |
| ☐ | ☒ | For null hypothesis testing, the test statistic (e.g. *F*, *t*, *r*) with confidence intervals, effect sizes, degrees of freedom and *P* value noted<br>*Give P values as exact values whenever suitable.* |
| ☒ | ☐ | For Bayesian analysis, information on the choice of priors and Markov chain Monte Carlo settings |
| ☒ | ☐ | For hierarchical and complex designs, identification of the appropriate level for tests and full reporting of outcomes |
| ☒ | ☐ | Estimates of effect sizes (e.g. Cohen's *d*, Pearson's *r*), indicating how they were calculated |

*Our web collection on statistics for biologists contains articles on many of the points above.*

## Software and code

Policy information about availability of computer code

| | |
|---|---|
| Data collection | Stool metagenome and metatranscriptome data was obtained from European Nucleotide Archives (v1.6.1) using 'enaBrowserTools' scripts that interface with the ENA servers, under accession number PRJNA389280 |
| Data analysis | FastQC (v0.11.7) and MultiQC (v.1.0) were used for quality checking metagenome and metatranscriptome data<br>BBTools (v38.51) (specifically BBDuk) was used for trimming and filtering<br>SortmeRNA (v4.3.3) was used to remove rRNA sequences from metatranscriptomic reads<br>DIAMOND (v2.0) was used for homology-based searches of metagenomes, metatranscriptomes, isolate genomes and isolate transcripts against both the publicly available HydDB dataset and an in-house database (available at https://doi.org/10.26180/c.5230745).<br>Filtering and normalisation of homology-based search results was conducted as per methods presented in Lappan et al (2023, Nature Microbiology), Bay et al. (Nature Microbiology 2021,) and Ortiz et al (2021, PNAS) among others.<br>CheckM (v 1.1.3) was used for isolate genome quality checks<br>GTDB-Tk (v1.6.0) was used for taxonomic characterisation of gut isolates and phylogenetic analysis<br>Prokka (v1.14.6) was used for isolate genome annotation<br>Salmon (v1.9.0) was used for transcript quantification<br>DRAM (v1.4.6) was used for transcriptome annotation with the KEGG protein database<br>AlphaFold2 (v2.1.1) was used for structural modelling, along with Foldseek and ChimeraX (v1.6.1)<br>TrimGalore (v0.6.6) was used for disease cohort metagenome quality control<br>Bowtie (v2.3.552) was used to remove reads that mapped to the human genome from disease cohort analysis<br>Seqtk (v1.3) was used to normalise for sequence depth for disease cohort metagenomes. |

For manuscripts utilizing custom algorithms or software that are central to the research but not yet described in published literature, software must be made available to editors and reviewers. We strongly encourage code deposition in a community repository (e.g. GitHub). See the Nature Portfolio guidelines for submitting code & software for further information.

## Data

Policy information about availability of data

All manuscripts must include a data availability statement. This statement should provide the following information, where applicable:
- Accession codes, unique identifiers, or web links for publicly available datasets
- A description of any restrictions on data availability
- For clinical datasets or third party data, please ensure that the statement adheres to our policy

> Stool metagenome and metatranscriptome datasets are available from ENA under accession number: PRJNA389280
> Biopsy metagenome data is available from ENA under accession number: PRJEB45397
> Isolate genomes are available from Australian Microbiome Culture Collection and ENA under accession numbers ERP105624 and ERP012217
> Gut isolate genomes and transcripts are available from ENA under accession number: PRJEB70412

## Research involving human participants, their data, or biological material

Policy information about studies with human participants or human data. See also policy information about sex, gender (identity/presentation), and sexual orientation and race, ethnicity and racism.

| | |
|---|---|
| Reporting on sex and gender | 54% of participants were female, primarily aged between 11-18 years of age. Considering this study was focused on microorganisms cultured from the biopsy samples, patient sex, gender and other demographics weren't analysed or controlled for. |
| Reporting on race, ethnicity, or other socially relevant groupings | Not applicable |
| Population characteristics | 42 pediatric patients with non-inflammatory gastrointestinal conditions. |
| Recruitment | Patients were recruited during pediatric endoscopy lists at Monash Children's Hospital from consenting participants receiving clinically indicated colonoscopies. |
| Ethics oversight | Human Research Ethics Committee (HREC) (HREC/16/MonH/253) and Monash 435 University Ethics Committee (Monash Health ref. 16367A |

Note that full information on the approval of the study protocol must also be provided in the manuscript.

# Field-specific reporting

Please select the one below that is the best fit for your research. If you are not sure, read the appropriate sections before making your selection.

☒ Life sciences ☐ Behavioural & social sciences ☐ Ecological, evolutionary & environmental sciences

For a reference copy of the document with all sections, see nature.com/documents/nr-reporting-summary-flat.pdf

# Life sciences study design

All studies must disclose on these points even when the disclosure is negative.

| | |
|---|---|
| Sample size | Sample size was not statistically predetermined and co-variates were not controlled for. The sample size of metagenomes, metatranscriptomes and isolate genomes was based on available, relevant datasets and what was considered representative of gastrointestinal microbiota. The sample size of cultured isolates was chosen based on ensuring taxonomic diversity, as well as including only those isolates that encoded the genes of interest. |
| Data exclusions | Six isolate genomes were excluded from analysis due to having high contamination (>10%) reported via CheckM. |
| Replication | Biological triplicates were used throughout the study for all experimental work, and all replication was successful. |
| Randomization | As this was a study involving genomic, biochemical and physiological analysis of microorganisms, randomisation was not necessary. |
| Blinding | As this was a study involving genomic, biochemical and physiological analysis of microorganisms, blinding was not necessary. |

# Reporting for specific materials, systems and methods

We require information from authors about some types of materials, experimental systems and methods used in many studies. Here, indicate whether each material, system or method listed is relevant to your study. If you are not sure if a list item applies to your research, read the appropriate section before selecting a response.

## Materials & experimental systems

| n/a | Involved in the study |
|-----|----------------------|
| ☒ | Antibodies |
| ☒ | Eukaryotic cell lines |
| ☒ | Palaeontology and archaeology |
| ☒ | Animals and other organisms |
| ☒ | Clinical data |
| ☒ | Dual use research of concern |
| ☒ | Plants |

## Methods

| n/a | Involved in the study |
|-----|----------------------|
| ☒ | ChIP-seq |
| ☒ | Flow cytometry |
| ☒ | MRI-based neuroimaging |

## Plants

| | |
|---|---|
| Seed stocks | *Report on the source of all seed stocks or other plant material used. If applicable, state the seed stock centre and catalogue number. If plant specimens were collected from the field, describe the collection location, date and sampling procedures.* |
| Novel plant genotypes | *Describe the methods by which all novel plant genotypes were produced. This includes those generated by transgenic approaches, gene editing, chemical/radiation-based mutagenesis and hybridization. For transgenic lines, describe the transformation method, the number of independent lines analyzed and the generation upon which experiments were performed. For gene-edited lines, describe the editor used, the endogenous sequence targeted for editing, the targeting guide RNA sequence (if applicable) and how the editor was applied.* |
| Authentication | *Describe any authentication procedures for each seed stock used or novel genotype generated. Describe any experiments used to assess the effect of a mutation and, where applicable, how potential secondary effects (e.g. second site T-DNA insertions, mosiacism, off-target gene editing) were examined.* |

