## [Peer Review File · Nature Microbiology]

A widespread hydrogenase supports fermentative growth of gut bacteria in healthy people

Corresponding Author: Professor Chris Greening

Version 0:

Reviewer comments:

Reviewer #1

(Remarks to the Author)

General points

A central claim of the manuscript is that “Bacteroides” and “group B [FeFe]-hydrogenases” are the major sources of H₂ in the gut. However, I do not find the presented data fully convincing. Based on the data presented in Figure S1, Bacteroides seem to produce an order of magnitude less H₂ than Clostridia that rely on A1 [FeFe]-hydrogenases. In addition, based on the presented data, it's unclear whether H₂ is a major or minor product of Bacteroides metabolism. Additional data to provide context about the levels of H₂ produced relative to glucose consumed or other fermentative products produced.

The authors conjecture at several points about the source and mechanism of H₂ production in Bacteroides. The conclusions seem plausible but would be much more convincing with additional experimental data. In particular, the authors posit the ferredoxin is likely the hydrogenase electron donor and that pyruvate-ferredoxin oxidoreductase may be the source of reduced ferredoxin. However, this is speculative based on limited data. Bacteroides genetics are relatively straightforward and it would be simple to test this hypothesis by generating pyruvate-ferredoxin oxidoreductase knockouts. It would also be useful to generate hydrogenase knockouts (though I do find the authors' conclusion about the responsible genes to be more compelling).

Gut Bacteroides require hemin added to the growth media for a functional fumarate reductase. With hemin provided, fumarate represents the major electron sink during fermentative growth. As this alternative electron sink could have a major impact on fermentative hydrogenase activity, the authors might consider assessing the effect of hemin on Bacteroides H₂ production.

Bacteroides have previously been shown to produce H₂ during fermentation. Citing papers that have shown this would be appropriate and could strengthen some of arguments, in my opinion.

Specific points

Title: I do not think the findings support the conclusion in the title that widespread hydrogenase “drives” fermentative growth. The authors show that hydrogenase activity is associated with fermentative growth but “drives” implies a far greater degree of essentiality than has been shown (or is likely in my opinion).

Abstract: The statement “we show that metabolically flexible respiratory bacteria are the most abundant H₂ oxidizers in the gut, not sulfate reducers, methanogens, and acetogens as previously thought” is a bit misleading, in my opinion. Really the contention has been that “sulfate reducers, methanogens, and acetogens” are dominant H₂ oxidizers (irrespective of their level of abundance) and the new analyses don't contradict that conclusion. The microbes identified by the authors as having flexible respiratory metabolisms use plenty of other oxidants than H₂ and so their higher abundance doesn't imply that they are more important oxidizers.

Line 172: I am not sure but, at least in *E. coli*, the periplasmic nitrate reductase (NapA) is not believed to be a respiratory enzyme and so I am not sure it makes sense to include it in the analysis.

Line 190: It seems premature to conclude that “group B [FeFe]-hydrogenase appears to drive fermentative H₂ production at this stage in the manuscript.

Line 231: I find the statement “Such *Clostridium* species appear to have evolved exceptionally rapid hydrogenases to enable vigorous growth in high nutrient conditions” unconvincing. These organisms express group A1 hydrogenases and this claim dismisses the observation that group A1 hydrogenases may be a more significant source of H₂. However, this possibility cannot be so easily dismissed. The “group B” expressing Bacteroides grow well too and are produce an order of magnitude less H₂ and

the importance of group A1 hydrogenases cannot be that easily ruled out.

Line 245: It would be helpful to have additional context to interpret the “0.68% and 2.50% levels of H₂.” How does this compare on a molar basis to the amount of glucose consumed or other fermentation products generated? Without such context, it’s unclear whether H₂ production reflects a major or minor activity.

Line 262: The conclusion in the section title and repeated in the section that “group B [FeFe]-hydrogenases to reoxidize ferredoxin” is not demonstrated by the data. The authors should either provide more data to support this conclusion or revise this section and acknowledge that the electron donor is unknown. One way to address this hypothesis would be to test a pyruvate-ferredoxin oxidoreductase KO strain for H₂ production.

Figure 3b: The authors should modify how this data is presented. There are so many zeros on the ppm axis that it’s challenging to compare the panels and recognize the substantial differences between strains in hydrogen production.

Line 315: The authors are proposing a hypothesis about H₂ production being part of the acetate production pathway, however, this is not experimentally addressed. Bacteroides are heme auxotrophs and heme is required for fumarate reduction. In the presence of heme, Bacteroides generally primarily ferment sugars to succinate/propionate. Thus, absent additional experimental data, this hypothesis does not seem particularly compelling.

Line 344/Figure S8: Two questions/points about these analyses 1) Are the differences reported here statistically significant? The data and conclusions about the effect on gnotobiotic mouse colonization seem particularly unconvincing – as there is clearly mouse-to-mouse variability in both directions. 2) These rb-tn-seq experiments are done in complex communities within chambers that contain high concentrations of H₂. It is thus possible that the hydrogenases are functioning in the reverse direction within this context. This may be worth considering.

Line 417: The argument that electron acceptors such as fumarate, nitrate, and sulfoxides might be important is based solely on the observation that these genes are abundant, however, gene abundance does not necessarily predict activity. Also, other compounds can serve as electron acceptors. The authors should either scale back this claim or expand the discussion to include the observation that other, perhaps unidentified, electron acceptors may be important.

Reviewer #2

(Remarks to the Author)

In this manuscript, Welsh et al investigate bacterial hydrogenases as key metabolic enzymes in the human gut microbiome. A survey stool metagenomes and a metatranscriptomics analysis reveal that group B [FeFe]-hydrogenases are more abundant and more highly transcribed than any other group of known hydrogenases. Uptake Group 1 [NiFe]-hydrogenases were found to be present and expressed as well, however, there was substantial inter-individual variation. Other hydrogenases were detected as well at lower levels. The authors next begin characterizing group B [FeFe]-hydrogenase activity in a number of diverse microorganisms under defined laboratory conditions. Gene expression of group B [FeFe]-hydrogenase correlated with hydrogenase activity, suggesting that this enzyme are likely the predominant hydrogenase. Notably, group B [FeFe]-hydrogenase activity is shown to occur in a number of Bacteroides isolates, suggesting that this genus might be a major producer of hydrogen in the large intestine. Structural prediction (AlphaFold2 modelling) suggests that the group B [FeFe]-hydrogenase from Bacteroides form monomers, with two distinct domains. A biochemical assay demonstrates bona fide hydrogenase activity. Metabolic modeling of different Bacteroides strains suggests that the group B [FeFe]-hydrogenase is involved in recycling ferredoxin. Transposon mutagenesis shows that mutants lacking the group B [FeFe]-hydrogenase were impaired for growth under fermentative conditions. An additional re-analysis of existing metagenomics data shows that altered abundance of various hydrogenases in Crohn’s disease patients, implying that altered hydrogen metabolism might be associated with Crohn’s disease.

While we know that microbial hydrogen metabolism is altered in a number of diseases, our molecular understanding of this type of metabolism is limited. This study makes several important, original observations: it reveals group B [FeFe]-hydrogenases (a poorly characterized group of hydrogenases) as the major molecular source, and Bacteroides as the likely main producers involved in release of molecular hydrogen (possibly to re-oxidize ferredoxin). This study also reveals new insights into hydrogen consumption, pointing to organisms with respiratory capacity as major consumer of hydrogen. Basic biochemical parameters and structural predictions are done on group B [FeFe]-hydrogenases, a previously uncharacterized group of hydrogenases. This information not only corrects prior assumptions of hydrogen metabolism (prior to this work, the assumption was that A1 hydrogenases [FeFe]-hydrogenases in Clostridia were major contributors), but provides a rationale to further investigate group B [FeFe]-hydrogenases.

I have no major concerns except that the authors may be over-interpreting some of the data of the modeling. For example, “AlphaFold2 modelling (Fig. S3 & S4) confirms group B [FeFe]-hydrogenases are structurally conserved between Bacteroides species (Fig. S5) and form monomers with two distinct globular domains”. I recommend that this should be toned down, or better supported by experimental data (for example that these enzymes are monomers).

Reviewer #3

(Remarks to the Author)

Although molecular hydrogen is an important fermentative product as well as an important substrate for some (pathogenic) prokaryotic members of the microbiome, the role of H₂ homeostasis, including the type of hydrogenases that produce most of the H₂ in the human gastrointestinal system, is poorly understood.

In a comprehensive and highly integrative approach, Greening and his co-workers first examined hundreds of human stool

metagenomes for the presence of hydrogenase genes of various types. They observed that the poorly characterized group B [FeFe]-hydrogenase genes are the most abundant, particularly in species of the Bacteroidetes phylum. Accompanying metatranscriptomic data showed that these genes are the most transcribed among the various hydrogenase genes, which was confirmed by the transcriptional analysis of seven isolated Bacteroides and four Clostridia species. Most of them produced H₂ in relatively high amounts during fermentative growth.

Since almost nothing is known about group B [FeFe]-hydrogenases, the authors performed AlphaFold2 modeling, which revealed a conserved structure of this monomeric enzyme in different Bacteroides species, including a chain of Fe-S clusters that presumably deliver electrons from reduced ferredoxin to the active site, and a typical H-cluster. In *E. coli* lysates in which the apo forms of the hydrogenases had been overproduced, the apo proteins could be activated by the addition of a chemically synthesized H-cluster mimic, and the resulting semi-synthetic hydrogenases (mostly) produced H₂ in considerable amounts from the artificial electron donor methyl viologen, reduced by dithionite. This showed that group B [FeFe]-hydrogenases, although they could not be purified, are indeed catalytically active.

To better understand the role of the group B [FeFe]-hydrogenases in their physiological context, the authors analyzed data sets from a random, barcoded transposon mutant library of *B. thetaiotaomicron* exposed to different (growth) conditions. This revealed that both hydrogenases of this organism are not essential for fermentative growth. Nevertheless, the absence of the group B [FeFe]-hydrogenase significantly reduced the fitness of the corresponding mutant in the presence of fermentable carbon sources, with glucose showing the strongest effect. This was consistent with the observation that this particular mutant showed the greatest fitness defect with the prodrug metronidazole, which becomes toxic when reduced by, e.g., reduced ferredoxin. This suggests that the mutant lacking group B [FeFe]-hydrogenase has increased levels of reduced ferredoxin, consistent with the proposed function of this enzyme.

Finally, it was found that the abundance of hydrogenase genes in the stool metagenomes of healthy and diseased people varied significantly. For example, the level of group B [FeFe]-hydrogenase genes was lower in patients with Crohn's disease, while the abundance of prototypical group A1 [FeFe]-hydrogenase genes was above the average level in healthy individuals. Therefore, these results could provide some clues for interpreting the phenotype of Crohn's disease.

In summary, this study contributes significantly to our understanding of the nature and role of hydrogenases in the human gut and provides a wealth of data that will serve as inspiration for future studies. The manuscript is very well written, referenced and the results are carefully interpreted. The content of the manuscript is definitely of very high interest to microbiologists, biochemists and also medical scientists. It is therefore very well suited to be published in *Nature Microbiology*.

I have only a few minor comments that should be considered before publication.

Page 5, lines 137-138: The by-sentence “, though operating alongside other enzymes” appears trivial. The authors may have meant “, though operating alongside other H₂-producing hydrogenases”, which would make sense.

Page 5, lines 145-146: The authors identify group B [FeFe]-hydrogenase as presumably most abundant hydrogenase in the human gut and this hydrogenase type is widespread in Bacteroides. To contextualize the above findings, the relative (average) abundance of Bacteroides species in the gut microbiome should be mentioned here and/or elsewhere.

Page 6, line 170 (and many other places!): The term “gene expression” encompasses both gene transcription and the translation of the corresponding mRNA into proteins. Thus, there are three levels (gene, transcript, protein) that are discussed in detail in this manuscript and should be named accordingly to avoid confusing the reader. In the example in line 170, the statement “enzymes except fumarate reductase” should be replaced by “genes except the genes encoding fumarate reductase” because transcriptome data are being discussed here. Another example of the need for more precise wording can be found on page 8, lines 238–240. Here, the genome of an actinobacterium carries a hydrogenase gene that is transcribed, but the transcript does not appear to be translated, i.e. no active enzyme is produced (or synthesized) because no H₂ has been observed. Therefore, it is inappropriate to claim that a group B [FeFe]-hydrogenase is expressed in this species, unless the authors define expression as mere gene transcription, which is currently not the case. This applies also for the titles of Figures 1-3. They should be “Abundance, expression, and distribution of hydrogenase genes and H₂-related metabolic genes throughout the human gut.”, “Phylogenomic tree showing distribution of hydrogenase genes among 812 bacterial isolates from the human gut.” and “Hydrogenase gene transcription and activity across 18 bacterial gut isolates.”, respectively. The authors should check the entire manuscript for correct wording in this regard.

Page 7, line 190: “..., much of which is likely to be partially recycled by respiratory...” (as mentioned in the introduction, H₂ gas is expelled as flatus or exhaled).

Page 7, line 212: It is not clear to the reader what is meant by “during fermentative survival”. If group 4a [NiFe]-hydrogenases are thought to be required for fermentative growth of these bacteria, this should be clearly stated.

Page 8, line 246: Another example for incorrect wording. “For all seven strains, the group B [FeFe]-hydrogenase genes were transcribed at high levels” (which indicates hydrogenase gene expression since H₂ was observed in the headspace of the cultures)

Page 9, line 279: The designation “His-ligated [4Fe-4S] cluster” is misleading, as it is coordinated by three cysteines and one histidine.

Page 9, line 279: “This ferredoxin-like domain is unusual in that its iron-sulfur clusters are modelled to be distant from the main body”. The structures presented in this study are based solely on AlphaFold2 calculations. Although the authors also used AF-Cluster to determine possible conformational changes, the proposed structures might be wrong. It is still only an assumption that the ferredoxin domain is beyond electron transfer distance to the corresponding [4Fe-4S]-cluster. It is important to mention the unusual conformation of the ferredoxin-like domain, but is not a hard fact. The authors could easily include ferredoxin in their AlphaFold2 calculations to support their assumption that the ferredoxin interaction enables productive electron transfer.

Page 9, line 287: “Trimeric enzymes” is a bit vague. Group A3 [FeFe]-hydrogenases usually form dimers of heterotrimers.

Page 9, lines 293-294: The authors demonstrate hydrogenase activity in cell lysates using dithionite-reduced methyl viologen as electron donor. The title of the chapter reads “Bacteroides use group B [FeFe]-hydrogenases to reoxidize ferredoxin during fermentation”. Although it is highly likely that ferredoxin is the native electron donor based on the knowledge accumulated in the literature and especially in this study, there is still no direct evidence. This should be taken into account.

Figure 4: This figure should only contain fields a-c. Panel d (EPR spectra) can be moved to the supplement and panel e should be an independent figure. Here, a seven-compartment box with the seven strains used for the transcriptional analysis should be presented in the figure. The legend (page 26) should explain the abbreviations of the different enzymes presented in the figure. Gray panels, as mentioned in the legend, are not present in the figure.

Page 11, lines 344-360: It should be better clarified that the barcoded transposon mutant library of *B. thetaiotaomiron* was already available. The observations that, e.g., transposon mutants were obtained for both hydrogenases currently sound as they were made by the authors of this manuscript. This also applies to the colonization experiments of germ-free mice (Figure S8F should read S8E). The negative effect of metronidazole on the fitness of group B [FeFe] hydrogenase gene mutants is quite clear and can be nicely explained. The same mutants appear to have a growth advantage in the presence of fusidic acid and NaF. The authors could comment on this as well.

Tables S1, S2, S4 and S7 (currently provided as Excel files with cryptic names) should be given unique names that should include S1, S2, S4 and S7. Furthermore, their content could be made more accessible with extended table legends.

Decision Letter:

13th December 2024

Dear Chris,

Thank you for your patience while your manuscript "A widespread hydrogenase drives fermentative growth of gut bacteria in healthy people" was under peer-review at Nature Microbiology. It has now been seen by 3 referees, whose expertise and comments you will find at the end of this email. Although they find your work of some potential interest, they have raised a number of concerns that will need to be addressed before we can consider publication of the work in Nature Microbiology.

In particular, all referees are enthusiastic about the paper but they ask that you tone down some conclusions in line with the data you have. Referee #1 also suggests some experiments to further support your findings that should be relatively straightforward to do.

Should further experimental data allow you to address these criticisms, we would be happy to look at a revised manuscript.

Please include a data availability statement as a separate section after Methods but before references, under the heading "Data Availability". This section should inform readers about the availability of the data used to support the conclusions of your study. This information includes accession codes to public repositories (data banks for protein, DNA or RNA sequences, microarray, proteomics data etc...), references to source data published alongside the paper, unique identifiers such as URLs to data repository entries, or data set DOIs, and any other statement about data availability. At a minimum, you should include the following statement: "The data that support the findings of this study are available from the corresponding author upon request", mentioning any restrictions on availability. If DOIs are provided, we also strongly encourage including these in the Reference list (authors, title, publisher (repository name), identifier, year). For more guidance on how to write this section please see: <http://www.nature.com/authors/policies/data/data-availability-statements-data-citations.pdf>

* If you have not done so already we suggest that you begin to revise your manuscript so that it conforms to our Article format instructions at <http://www.nature.com/nmicrobiol/info/final-submission>. Refer also to any guidelines provided in this letter.

When submitting the revised version of your manuscript, please pay close attention to our [href="https://www.nature.com/nature-portfolio/editorial-policies/image-integrity">Digital Image Integrity Guidelines](https://www.nature.com/nature-portfolio/editorial-policies/image-integrity) and to the following points below:

Link Redacted

Note: This url links to your confidential homepage and associated information about manuscripts you may have submitted or be reviewing for us. If you wish to forward this e-mail to co-authors, please delete this link to your homepage first.

Nature Microbiology is committed to improving transparency in authorship. As part of our efforts in this direction, we are now requesting that all authors identified as 'corresponding author' on published papers create and link their Open Researcher and Contributor Identifier (ORCID) with their account on the Manuscript Tracking System (MTS), prior to acceptance. This applies to primary research papers only. ORCID helps the scientific community achieve unambiguous attribution of all scholarly contributions. You can create and link your ORCID from the home page of the MTS by clicking on 'Modify my Springer Nature account'. For more information please visit www.springernature.com/orcid.

If you wish to submit a suitably revised manuscript we would hope to receive it within 6 months. If you cannot send it within this time, please let us know. We will be happy to consider your revision, even if a similar study has been accepted for publication at Nature Microbiology or published elsewhere (up to a maximum of 6 months).

Yours sincerely,

Reviewer Expertise:

Referee #1: microbial metabolism
Referee #2: microbial metabolism
Referee #3: microbial hydrogenases

Reviewer Comments:

Reviewer #1 (Remarks to the Author):

General points

A central claim of the manuscript is that "Bacteroides" and "group B [FeFe]-hydrogenases" are the major sources of H₂ in the gut. However, I do not find the presented data fully convincing. Based on the data presented in Figure S1, Bacteroides seem to produce an order of magnitude less H₂ than Clostridia that rely on A1 [FeFe]-hydrogenases. In addition, based on the presented data, it's unclear whether H₂ is a major or minor product of Bacteroides metabolism. Additional data to provide context about the levels of H₂ produced relative to glucose consumed or other fermentative products produced.

The authors conjecture at several points about the source and mechanism of H₂ production in Bacteroides. The conclusions seem plausible but would be much more convincing with additional experimental data. In particular, the authors posit the ferredoxin is likely the hydrogenase electron donor and that pyruvate-ferredoxin oxidoreductase may be the source of reduced ferredoxin. However, this is speculative based on limited data. Bacteroides genetics are relatively straightforward and it would be simple to test this hypothesis by generating pyruvate-ferredoxin oxidoreductase knockouts. It would also be useful to generate hydrogenase knockouts (though I do find the authors' conclusion about the responsible genes to be more compelling).

Gut Bacteroides require hemin added to the growth media for a functional fumarate reductase. With hemin provided, fumarate represents the major electron sink during fermentative growth. As this alternative electron sink could have a major impact on fermentative hydrogenase activity, the authors might consider assessing the effect of hemin on Bacteroides H₂ production.

Bacteroides have previously been shown to produce H₂ during fermentation. Citing papers that have shown this would be appropriate and could strengthen some of arguments, in my opinion.

Specific points

Title: I do not think the findings support the conclusion in the title that widespread hydrogenase "drives" fermentative growth. The authors show that hydrogenase activity is associated with fermentative growth but "drives" implies a far greater degree of essentiality than has been shown (or is likely in my opinion).

Abstract: The statement "we show that metabolically flexible respiratory bacteria are the most abundant H₂ oxidizers in the gut, not sulfate reducers, methanogens, and acetogens as previously thought" is a bit misleading, in my opinion. Really the contention has been that "sulfate reducers, methanogens, and acetogens" are dominant H₂ oxidizers (irrespective of their level of abundance) and the new analyses don't contradict that conclusion. The microbes identified by the authors as having flexible

respiratory metabolisms use plenty of other oxidants than H₂ and so their higher abundance doesn't imply that they are more important oxidizers.

Line 172: I am not sure but, at least in *E. coli*, the periplasmic nitrate reductase (NapA) is not believed to be a respiratory enzyme and so I am not sure it makes sense to include it in the analysis.

Line 190: It seems premature to conclude that "group B [FeFe]-hydrogenase appears to drive fermentative H₂ production at this stage in the manuscript.

Line 231: I find the statement "Such *Clostridium* species appear to have evolved exceptionally rapid hydrogenases to enable vigorous growth in high nutrient conditions" unconvincing. These organisms express group A1 hydrogenases and this claim dismisses the observation that group A1 hydrogenases may be a more significant source of H₂. However, this possibility cannot be so easily dismissed. The "group B" expressing *Bacteroides* grow well too and are produce an order of magnitude less H₂ and the importance of group A1 hydrogenases cannot be that easily ruled out.

Line 245: It would be helpful to have additional context to interpret the "0.68% and 2.50% levels of H₂." How does this compare on a molar basis to the amount of glucose consumed or other fermentation products generated? Without such context, it's unclear whether H₂ production reflects a major or minor activity.

Line 262: The conclusion in the section title and repeated in the section that "group B [FeFe]-hydrogenases to reoxidize ferredoxin" is not demonstrated by the data. The authors should either provide more data to support this conclusion or revise this section and acknowledge that the electron donor is unknown. One way to address this hypothesis would be to test a pyruvate-ferredoxin oxidoreductase KO strain for H₂ production.

Figure 3b: The authors should modify how this data is presented. There are so many zeros on the ppm axis that it's challenging to compare the panels and recognize the substantial differences between strains in hydrogen production.

Line 315: The authors are proposing a hypothesis about H₂ production being part of the acetate production pathway, however, this is not experimentally addressed. *Bacteroides* are heme auxotrophs and heme is required for fumarate reduction. In the presence of heme, *Bacteroides* generally primarily ferment sugars to succinate/propionate. Thus, absent additional experimental data, this hypothesis does not seem particularly compelling.

Line 344/Figure S8: Two questions/points about these analyses 1) Are the differences reported here statistically significant? The data and conclusions about the effect on gnotobiotic mouse colonization seem particularly unconvincing – as there is clearly mouse-to-mouse variability in both directions. 2) These rb-tn-seq experiments are done in complex communities within chambers that contain high concentrations of H₂. It is thus possible that the hydrogenases are functioning in the reverse direction within this context. This may be worth considering.

Line 417: The argument that electron acceptors such as fumarate, nitrate, and sulfoxides might be important is based solely on the observation that these genes are abundant, however, gene abundance does not necessarily predict activity. Also, other compounds can serve as electron acceptors. The authors should either scale back this claim or expand the discussion to include the observation that other, perhaps unidentified, electron acceptors may be important.

Reviewer #2 (Remarks to the Author):

In this manuscript, Welsh et al investigate bacterial hydrogenases as key metabolic enzymes in the human gut microbiome. A survey stool metagenomes and a metatranscriptomics analysis reveal that group B [FeFe]-hydrogenases are more abundant and more highly transcribed than any other group of known hydrogenases. Uptake Group 1 [NiFe]-hydrogenases were found to be present and expressed as well, however, there was substantial inter-individual variation. Other hydrogenases were detected as well at lower levels. The authors next begin characterizing group B [FeFe]-hydrogenase activity in a number of diverse microorganisms under defined laboratory conditions. Gene expression of group B [FeFe]-hydrogenase correlated with hydrogenase activity, suggesting that this enzyme are likely the predominant hydrogenase. Notably, group B [FeFe]-hydrogenase activity is shown to occur in a number of *Bacteroides* isolates, suggesting that this genus might be a major producer of hydrogen in the large intestine. Structural prediction (AlphaFold2 modelling) suggests that the group B [FeFe]-hydrogenase from *Bacteroides* form monomers, with two distinct domains. A biochemical assay demonstrates bona fide hydrogenase activity. Metabolic modeling of different *Bacteroides* strains suggests that the group B [FeFe]-hydrogenase is involved in recycling ferredoxin. Transposon mutagenesis shows that mutants lacking the group B [FeFe]-hydrogenase were impaired for growth under fermentative conditions. An additional re-analysis of existing metagenomics data shows that altered abundance of various hydrogenases in Crohn's disease patients, implying that altered hydrogen metabolism might be associated with Crohn's disease.

While we know that microbial hydrogen metabolism is altered in a number of diseases, our molecular understanding of this type of metabolism is limited. This study makes several important, original observations: it reveals group B [FeFe]-hydrogenases (a poorly characterized group of hydrogenases) as the major molecular source, and *Bacteroides* as the likely main producers involved in release of molecular hydrogen (possibly to re-oxidize ferredoxin). This study also reveals new insights into hydrogen consumption, pointing to organisms with respiratory capacity as major consumer of hydrogen. Basic biochemical parameters and structural predictions are done on group B [FeFe]-hydrogenases, a previously uncharacterized group of hydrogenases. This information not only corrects prior assumptions of hydrogen metabolism (prior to this work, the assumption was that A1 hydrogenases [FeFe]-hydrogenases in *Clostridia* were major contributors), but provides a rationale to further investigate group B [FeFe]-hydrogenases.

I have no major concerns except that the authors may be over-interpreting some of the data of the modeling. For example, "AlphaFold2 modelling (Fig. S3 & S4) confirms group B [FeFe]-hydrogenases are structurally conserved between Bacteroides species (Fig. S5) and form monomers with two distinct globular domains". I recommend that this should be toned down, or better supported by experimental data (for example that these enzymes are monomers).

Reviewer #3 (Remarks to the Author):

Although molecular hydrogen is an important fermentative product as well as an important substrate for some (pathogenic) prokaryotic members of the microbiome, the role of H₂ homeostasis, including the type of hydrogenases that produce most of the H₂ in the human gastrointestinal system, is poorly understood.

In a comprehensive and highly integrative approach, Greening and his co-workers first examined hundreds of human stool metagenomes for the presence of hydrogenase genes of various types. They observed that the poorly characterized group B [FeFe]-hydrogenase genes are the most abundant, particularly in species of the Bacteroidetes phylum. Accompanying metatranscriptomic data showed that these genes are the most transcribed among the various hydrogenase genes, which was confirmed by the transcriptional analysis of seven isolated Bacteroides and four Clostridia species. Most of them produced H₂ in relatively high amounts during fermentative growth.

Since almost nothing is known about group B [FeFe]-hydrogenases, the authors performed AlphaFold2 modeling, which revealed a conserved structure of this monomeric enzyme in different Bacteroides species, including a chain of Fe-S clusters that presumably deliver electrons from reduced ferredoxin to the active site, and a typical H-cluster. In E. coli lysates in which the apo forms of the hydrogenases had been overproduced, the apo proteins could be activated by the addition of a chemically synthesized H-cluster mimic, and the resulting semi-synthetic hydrogenases (mostly) produced H₂ in considerable amounts from the artificial electron donor methyl viologen, reduced by dithionite. This showed that group B [FeFe]-hydrogenases, although they could not be purified, are indeed catalytically active.

To better understand the role of the group B [FeFe]-hydrogenases in their physiological context, the authors analyzed data sets from a random, barcoded transposon mutant library of B. thetaiotaomicron exposed to different (growth) conditions. This revealed that both hydrogenases of this organism are not essential for fermentative growth. Nevertheless, the absence of the group B [FeFe]-hydrogenase significantly reduced the fitness of the corresponding mutant in the presence of fermentable carbon sources, with glucose showing the strongest effect. This was consistent with the observation that this particular mutant showed the greatest fitness defect with the prodrug metronidazole, which becomes toxic when reduced by, e.g., reduced ferredoxin. This suggests that the mutant lacking group B [FeFe]-hydrogenase has increased levels of reduced ferredoxin, consistent with the proposed function of this enzyme.

Finally, it was found that the abundance of hydrogenase genes in the stool metagenomes of healthy and diseased people varied significantly. For example, the level of group B [FeFe]-hydrogenase genes was lower in patients with Crohn's disease, while the abundance of prototypical group A1 [FeFe]-hydrogenase genes was above the average level in healthy individuals. Therefore, these results could provide some clues for interpreting the phenotype of Crohn's disease.

In summary, this study contributes significantly to our understanding of the nature and role of hydrogenases in the human gut and provides a wealth of data that will serve as inspiration for future studies. The manuscript is very well written, referenced and the results are carefully interpreted. The content of the manuscript is definitely of very high interest to microbiologists, biochemists and also medical scientists. It is therefore very well suited to be published in Nature Microbiology.

I have only a few minor comments that should be considered before publication.

Page 5, lines 137-138: The by-sentence " , though operating alongside other enzymes" appears trivial. The authors may have meant " , though operating alongside other H₂-producing hydrogenases", which would make sense.

Page 5, lines 145-146: The authors identify group B [FeFe]-hydrogenase as presumably most abundant hydrogenase in the human gut and this hydrogenase type is widespread in Bacteroides. To contextualize the above findings, the relative (average) abundance of Bacteroides species in the gut microbiome should be mentioned here and/or elsewhere.

Page 6, line 170 (and many other places!): The term "gene expression" encompasses both gene transcription and the translation of the corresponding mRNA into proteins. Thus, there are three levels (gene, transcript, protein) that are discussed in detail in this manuscript and should be named accordingly to avoid confusing the reader. In the example in line 170, the statement "enzymes except fumarate reductase" should be replaced by "genes except the genes encoding fumarate reductase" because transcriptome data are being discussed here. Another example of the need for more precise wording can be found on page 8, lines 238-240. Here, the genome of an actinobacterium carries a hydrogenase gene that is transcribed, but the transcript does not appear to be translated, i.e. no active enzyme is produced (or synthesized) because no H₂ has been observed. Therefore, it is inappropriate to claim that a group B [FeFe]-hydrogenase is expressed in this species, unless the authors define expression as mere gene transcription, which is currently not the case. This applies also for the titles of Figures 1-3. They should be "Abundance, expression, and distribution of hydrogenase genes and H₂-related metabolic genes throughout the human gut.", "Phylogenomic tree showing distribution of hydrogenase genes among 812 bacterial isolates from the human gut." and "Hydrogenase gene transcription and activity across 18 bacterial gut isolates.", respectively. The authors should check the entire manuscript for correct wording in this regard.

Page 7, line 190: "... , much of which is likely to be partially recycled by respiratory..." (as mentioned in the introduction, H₂ gas is expelled as flatus or exhaled).

Page 7, line 212: It is not clear to the reader what is meant by "during fermentative survival". If group 4a [NiFe]-hydrogenases are thought to be required for fermentative growth of these bacteria, this should be clearly stated.

Page 8, line 246: Another example for incorrect wording. "For all seven strains, the group B [FeFe]-hydrogenase genes were transcribed at high levels" (which indicates hydrogenase gene expression since H₂ was observed in the headspace of the cultures)

Page 9, line 279: The designation "His-ligated [4Fe-4S] cluster" is misleading, as it is coordinated by three cysteines and one histidine.

Page 9, line 279: "This ferredoxin-like domain is unusual in that its iron-sulfur clusters are modelled to be distant from the main body". The structures presented in this study are based solely on AlphaFold2 calculations. Although the authors also used AF-

Cluster to determine possible conformational changes, the proposed structures might be wrong. It is still only an assumption that the ferredoxin domain is beyond electron transfer distance to the corresponding [4Fe-4S]-cluster. It is important to mention the unusual conformation of the ferredoxin-like domain, but is not a hard fact. The authors could easily include ferredoxin in their AlphaFold2 calculations to support their assumption that the ferredoxin interaction enables productive electron transfer.

Page 9, line 287: "Trimic enzymes" is a bit vague. Group A3 [FeFe]-hydrogenases usually form dimers of heterotrimers.

Page 9, lines 293-294: The authors demonstrate hydrogenase activity in cell lysates using dithionite-reduced methyl viologen as electron donor. The title of the chapter reads "Bacteroides use group B [FeFe]-hydrogenases to reoxidize ferredoxin during fermentation". Although it is highly likely that ferredoxin is the native electron donor based on the knowledge accumulated in the literature and especially in this study, there is still no direct evidence. This should be taken into account.

Figure 4: This figure should only contain fields a-c. Panel d (EPR spectra) can be moved to the supplement and panel e should be an independent figure. Here, a seven-compartment box with the seven strains used for the transcriptional analysis should be presented in the figure. The legend (page 26) should explain the abbreviations of the different enzymes presented in the figure. Gray panels, as mentioned in the legend, are not present in the figure.

Page 11, lines 344-360: It should be better clarified that the barcoded transposon mutant library of *B. thetaiotaomiron* was already available. The observations that, e.g., transposon mutants were obtained for both hydrogenases currently sound as they were made by the authors of this manuscript. This also applies to the colonization experiments of germ-free mice (Figure S8F should read S8E). The negative effect of metronidazole on the fitness of group B [FeFe] hydrogenase gene mutants is quite clear and can be nicely explained. The same mutants appear to have a growth advantage in the presence of fusidic acid and NaF. The authors could comment on this as well.

Tables S1, S2, S4 and S7 (currently provided as Excel files with cryptic names) should be given unique names that should include S1, S2, S4 and S7. Furthermore, their content could be made more accessible with extended table legends.

Version 1:

Reviewer comments:

Reviewer #1

(Remarks to the Author)

The authors have made substantial efforts to address the concerns raised in the initial review, and the revised manuscript shows significant improvement through additional experimental work and methodological refinements. In particular, the biochemical validation experiments demonstrating PFOR-dependent H₂ production and the hemin-dependent physiological studies represent valuable additions that strengthen the mechanistic claims.

However, one fundamental issue could be better addressed. While the authors conducted additional experiments in response to my concerns about quantitative significance, they failed to provide clear stoichiometric analysis and thus whether H₂ represents a major or minor metabolic flux in *Bacteroides* remains unclear. Figure S9 shows glucose consumption and H₂ production on different scales and units (mg/100ml glucose vs. ppm H₂), making their claim "H₂ production is approximately proportional to glucose consumption and growth" impossible to evaluate. A simple calculation showing "H₂ accounts for X% of carbon flux" would directly address this question. Without this quantitative framework, readers cannot assess whether the group B [FeFe]-hydrogenase represents a metabolically significant pathway or a minor side reaction, undermining one of the paper's central claims about the importance of this enzyme in gut metabolism.

Reviewer #2

(Remarks to the Author)

The authors have adequately addressed my concerns.

Reviewer #3

(Remarks to the Author)

The authors have taken my comments and criticisms into account in an excellent manner. I also believe that the criticisms raised by the other reviewers have been adequately addressed. With regard to the terminology used in the manuscript, I recommend checking the text again for consistency. For example, genes encode proteins (in most cases), but they are not themselves encoded, as mentioned, e.g., in lines 135 – 136.

Further suggested changes in this regard:

Line 128: "...metatranscriptomes confirm that these hydrogenase genes are highly..."

Line 370: "...lactate dehydrogenase gene varied..."

Line 375: "...during fermentative growth, the corresponding gene remains transcribed ..."

Decision Letter:

7th July 2025

Dear Professor Greening,

Thank you for your patience while your manuscript "A widespread hydrogenase drives fermentative growth of gut bacteria in healthy people" was under peer-review at Nature Microbiology. It has now been seen by 3 referees, whose comments you will find at the of this email. You will see from their comments below that while majority of their concerns are addressed, R1 still feels that it is not clear if H2 represents a major or a minor metabolic flux in Bacteroides. They have suggested to perform stoichiometric analysis to support the claimed relationship between glucose consumption and H2 production. Similar to the referee, we also feel that this will significantly strengthen the central claim of the manuscript. We are very interested in the possibility of publishing your study in Nature Microbiology, but would like to consider your response in the form of a revised manuscript before we make a final decision on publication.

If you have not done so already please begin to revise your manuscript so that it conforms to our Article format instructions at <http://www.nature.com/nmicrobiol/info/final-submission/>

The usual length limit for a Nature Microbiology Article is six display items (figures or tables) and 3,000 words. We have some flexibility, and can allow a revised manuscript at 3,500 words, but please consider this a firm upper limit. There is a trade-off of ~250 words per display item, so if you need more space, you could move a Figure or Table to Supplementary Information.

Some reduction could be achieved by focusing any introductory material and moving it to the start of your opening 'bold' paragraph, whose function is to outline the background to your work, describe in a sentence your new observations, and explain your main conclusions. The discussion should also be limited. Methods should be described in a separate section following the discussion, we do not place a word limit on Methods.

Nature Microbiology titles should give a sense of the main new findings of a manuscript, and should not contain punctuation. Please keep in mind that we strongly discourage active verbs in titles, and that they should ideally fit within 90 characters each (including spaces).

Please include a data availability statement as a separate section after Methods but before references, under the heading "Data Availability". This section should inform readers about the availability of the data used to support the conclusions of your study. This information includes accession codes to public repositories (data banks for protein, DNA or RNA sequences, microarray, proteomics data etc...), references to source data published alongside the paper, unique identifiers such as URLs to data repository entries, or data set DOIs, and any other statement about data availability. At a minimum, you should include the following statement: "The data that support the findings of this study are available from the corresponding author upon request", mentioning any restrictions on availability. If DOIs are provided, we also strongly encourage including these in the Reference list (authors, title, publisher (repository name), identifier, year). For more guidance on how to write this section please see: <http://www.nature.com/authors/policies/data/data-availability-statements-data-citations.pdf>

To improve the accessibility of your paper to readers from other research areas, please pay particular attention to the wording of the paper's opening bold paragraph, which serves both as an introduction and as a brief, non-technical summary in about 150 words. If, however, you require one or two extra sentences to explain your work clearly, please include them even if the paragraph is over-length as a result. The opening paragraph should not contain references. Because scientists from other sub-disciplines will be interested in your results and their implications, it is important to explain essential but specialised terms concisely. We suggest you show your summary paragraph to colleagues in other fields to uncover any problematic concepts.

If your paper is accepted for publication, we will edit your display items electronically so they conform to our house style and will reproduce clearly in print. If necessary, we will re-size figures to fit single or double column width. If your figures contain several parts, the parts should form a neat rectangle when assembled. Choosing the right electronic format at this stage will speed up the processing of your paper and give the best possible results in print. We would like the figures to be supplied as vector files - EPS, PDF, AI or postscript (PS) file formats (not raster or bitmap files), preferably generated with vector-graphics software (Adobe Illustrator for example). Please try to ensure that all figures are non-flattened and fully editable. All images should be at least 300 dpi resolution (when figures are scaled to approximately the size that they are to be printed at) and in RGB colour format. Please do not submit Jpeg or flattened TIFF files. Please see also 'Guidelines for Electronic Submission of Figures' at the end of this letter for further detail.

Figure legends must provide a brief description of the figure and the symbols used, within 350 words, including definitions of any error bars employed in the figures.

When submitting the revised version of your manuscript, please pay close attention to our href="https://www.nature.com/nature-

research/editorial-policies/image-integrity">Digital Image Integrity Guidelines. and to the following points below:

EXTENDED DATA FIGURES

Please include a statement before the acknowledgements naming the author to whom correspondence and requests for materials should be addressed.

Finally, we require authors to include a statement of their individual contributions to the paper -- such as experimental work, project planning, data analysis, etc. -- immediately after the acknowledgements. The statement should be short, and refer to authors by their initials. For details please see the Authorship section of our joint Editorial policies at http://www.nature.com/authors/editorial_policies/authorship.html

* include a point-by-point response to any editorial suggestions and to our referees. Please include your response to the editorial suggestions in your cover letter, and please upload your response to the referees as a separate document.

* ensure it complies with our format requirements for Letters as set out in our guide to authors at www.nature.com/nmicrobiol/info/gta/

* state in a cover note the length of the text, methods and legends; the number of references; number and estimated final size of figures and tables

* resubmit electronically if possible using the link below to access your home page:

Link Redacted

*This url links to your confidential homepage and associated information about manuscripts you may have submitted or be reviewing for us. If you wish to forward this e-mail to co-authors, please delete this link to your homepage first.

Please ensure that all correspondence is marked with your Nature Microbiology reference number in the subject line.

Nature Microbiology is committed to improving transparency in authorship. As part of our efforts in this direction, we are now requesting that all authors identified as 'corresponding author' on published papers create and link their Open Researcher and Contributor Identifier (ORCID) with their account on the Manuscript Tracking System (MTS), prior to acceptance. This applies to primary research papers only. ORCID helps the scientific community achieve unambiguous attribution of all scholarly contributions. You can create and link your ORCID from the home page of the MTS by clicking on 'Modify my Springer Nature account'. For more information please visit www.springernature.com/orcid.

We hope to receive your revised paper within three weeks. If you cannot send it within this time, please let us know.

Yours sincerely,

Reviewers Comments:

Reviewer #1 (Remarks to the Author):

The authors have made substantial efforts to address the concerns raised in the initial review, and the revised manuscript shows significant improvement through additional experimental work and methodological refinements. In particular, the biochemical validation experiments demonstrating PFOR-dependent H₂ production and the hemin-dependent physiological studies represent valuable additions that strengthen the mechanistic claims.

However, one fundamental issue could be better addressed. While the authors conducted additional experiments in response to my concerns about quantitative significance, they failed to provide clear stoichiometric analysis and thus whether H₂ represents a major or minor metabolic flux in *Bacteroides* remains unclear. Figure S9 shows glucose consumption and H₂ production on different scales and units (mg/100ml glucose vs. ppm H₂), making their claim “H₂ production is approximately proportional to glucose consumption and growth” impossible to evaluate. A simple calculation showing “H₂ accounts for X% of carbon flux” would directly address this question. Without this quantitative framework, readers cannot assess whether the group B [FeFe]-hydrogenase represents a metabolically significant pathway or a minor side reaction, undermining one of the paper's central claims about the importance of this enzyme in gut metabolism.

Reviewer #2 (Remarks to the Author):

The authors have adequately addressed my concerns.

Reviewer #3 (Remarks to the Author):

The authors have taken my comments and criticisms into account in an excellent manner. I also believe that the criticisms raised by the other reviewers have been adequately addressed. With regard to the terminology used in the manuscript, I recommend checking the text again for consistency. For example, genes encode proteins (in most cases), but they are not themselves encoded, as mentioned, e.g., in lines 135 – 136.

Further suggested changes in this regard:

Line 128: “...metatranscriptomes confirm that these hydrogenase genes are highly...”

Line 370: “...lactate dehydrogenase gene varied...”

Line 375: “...during fermentative growth, the corresponding gene remains transcribed ...”

Version 2:

Reviewer comments:

Reviewer #1

(Remarks to the Author)

The authors have addressed my questions.

Decision Letter:

Our ref: NMICROBIOL-24103301B

25th July 2025

Dear Dr. Greening,

Thank you for submitting your revised manuscript "A widespread hydrogenase drives fermentative growth of gut bacteria in healthy people" (NMICROBIOL-24103301B). It has now been seen by the original referees and their comments are below. The reviewers find that the paper has improved in revision, and therefore we'll be happy to publish it, in principle, in *Nature Microbiology*, pending minor revisions to satisfy the referees' final requests and to comply with our editorial and formatting guidelines.

Thank you again for your interest in *Nature Microbiology*. Please do not hesitate to contact me if you have any questions.

Sincerely,

Reviewer #1 (Remarks to the Author):

The authors have addressed my questions.

Version 3:

Decision Letter:

16th September 2025

Dear Chris and Cait,

I am pleased to accept your Article "A widespread hydrogenase supports fermentative growth of gut bacteria in healthy people" for publication in Nature Microbiology. Thank you for having chosen to submit your work to us and many congratulations.

Authors may need to take specific actions to achieve compliance with funder and institutional open access mandates. If your research is supported by a funder that requires immediate open access (e.g. according to [a Plan S principles](https://www.springernature.com/gp/open-science/plan-s-compliance) or the [NIH public access policy](https://www.springernature.com/gp/open-science/us-federal-agency-compliance)) then you should select the gold OA route, and we will direct you to the compliant route where possible. Because authors warrant under our subscription licensing terms that they haven't committed to licensing any version of their article under a licence inconsistent with the terms of our agreement – including the applicable embargo period – publication under the subscription model isn't suitable for authors whose funders require no embargo.

You can now use a single sign-on for all your accounts, view the status of all your manuscript submissions and reviews, access

usage statistics for your published articles and download a record of your refereeing activity for the Nature journals.

Best wishes,

P.S. Click on the following link if you would like to recommend Nature Microbiology to your librarian
<http://www.nature.com/subscriptions/recommend.html#forms>

** Visit the Springer Nature Editorial and Publishing website at http://editorial-jobs.springernature.com?utm_source=ejP_NMicro_email&utm_medium=ejP_NMicro_email&utm_campaign=ejp_NMicro for more information about our career opportunities. If you have any questions please click [here](mailto:editorial.publishing.jobs@springernature.com).**

Response To Reviewers

Reviewer #1

Reviewer concerns about the significance of H₂ production in overall Bacteroides metabolism, particularly the role of Heme.

- 1. A central claim of the manuscript is that “Bacteroides” and “group B [FeFe]-hydrogenases” are the major sources of H₂ in the gut. However, I do not find the presented data fully convincing. Based on the data presented in Figure S1, Bacteroides seem to produce an order of magnitude less H₂ than Clostridia that rely on A1 [FeFe]-hydrogenases. In addition, based on the presented data, it’s unclear whether H₂ is a major or minor product of Bacteroides metabolism. Additional data to provide context about the levels of H₂ produced relative to glucose consumed or other fermentative products produced.**
- 2. Line 245: It would be helpful to have additional context to interpret the “0.68% and 2.50% levels of H₂.” How does this compare on a molar basis to the amount of glucose consumed or other fermentation products generated? Without such context, it’s unclear whether H₂ production reflects a major or minor activity.**
- 3. Gut Bacteroides require hemin added to the growth media for a functional fumarate reductase. With hemin provided, fumarate represents the major electron sink during fermentative growth. As this alternative electron sink could have a major impact on fermentative hydrogenase activity, the authors might consider assessing the effect of hemin on Bacteroides H₂ production**
- 4. Line 315: The authors are proposing a hypothesis about H₂ production being part of the acetate production pathway, however, this is not experimentally addressed. Bacteroides are heme auxotrophs and heme is required for fumarate reduction. In the presence of heme, Bacteroides generally primarily ferment sugars to succinate/propionate. Thus, absent additional experimental data, this hypothesis does not seem particularly compelling.**

We thank the reviewer for these insightful comments and raising this critical point regarding the importance of H₂ production in overall *Bacteroides* metabolism. To further contextualise the relationship between H₂ production, glucose fermentation and hemin-dependent respiration, we conducted additional experiments using a defined medium previously used for studies investigating *Bacteroides* carbohydrate metabolism¹ (mBMM, modified *Bacteroides* minimal media). We measured *Bacteroides fragilis* growth, glucose consumption, VFA profiles and H₂ production under varying glucose concentrations, and with or without hemin. As the reviewer suggests, hemin is required for functional fumarate reductase activity in *Bacteroides*, enabling respiration via fumarate as a terminal electron acceptor. We found that H₂ production, when normalised to growth, was ~19% higher in the absence of hemin (Fig. R1, in-text Fig. 4c), indicating that when the fumarate reductase is not active, the cells shift towards a more fermentative metabolism with increased reliance on H₂ production to dispose of excess reductant. Conversely, under hemin availability, cells likely direct electron flow towards fumarate reduction, which competes with the hydrogenase as an electron sink, decreasing H₂ production. However, substantial H₂ was still produced, even in the presence of hemin,

supporting the idea that H₂ production is central to the metabolism of *Bacteroides*, which is consistent with the expression of the group B [FeFe]-hydrogenase under the original YCFA condition (which includes hemin). These results are now included in the revised manuscript and discussed in lines 343-358.

Additionally, we have clarified that H₂ production is approximately proportional to glucose consumption and growth (Fig. R2, in-text Fig. S9), which is consistent with its role as a central product of *Bacteroides* fermentative metabolism. To further contextualise H₂ production, we analysed the production of fermentation end-products by HPLC, under both the presence and absence of hemin. In the presence of hemin, cultures produced high-concentrations of succinate, along with acetate and propionate, which is consistent with hemin-dependent fumarate reduction and succinate-propionate pathway (Fig. R3, in-text Fig. S10). Nonetheless, substantial amounts of H₂ was still produced from glucose consumption even in the presence of hemin.

1 Qu, Z. *et al.* Selective utilization of medicinal polysaccharides by human gut *Bacteroides* and *Parabacteroides* species. *Nature Communications* **16**, 638 (2025).

Figure R1. Comparisons of *Bacteroides fragilis* H₂ production (ppm) per unit of biomass increase (OD₆₀₀) in the presence of heme (pink line) and absence of heme (green line).

Figure R2. Growth (green line, first right y axis), glucose consumption (pink line, left y axis) and H₂ production (purple bars, second right y axis) observed for triplicate cultures of *Bacteroides fragilis* over 96 hours in media containing **a)** high and **b)** lower concentrations of glucose.

Figure R3. Volatile fatty acid (VFA) measurements (mM) relative to growth (OD₆₀₀) of *B. fragilis* culture supernatant under the presence (+hemin, pink) or absence (-hemin, purple) of hemin (n=3). Statistical significance was assessed using an unpaired t-test with Welch's correction, which does not assume equal variances between groups (*, P < 0.05; **, P < 0.01; ***, P < 0.001; ****, P < 0.0001). Error bars represent the standard deviation from the mean.

Reviewer concerns about whether the group B [FeFe]-hydrogenase is involved in reoxidation of reduced ferredoxin provided by the pyruvate-ferredoxin oxidoreductase.

5. **The authors conjecture at several points about the source and mechanism of H₂ production in *Bacteroides*. The conclusions seem plausible but would be much more convincing with additional experimental data. In particular, the authors posit the ferredoxin is likely the hydrogenase electron donor and that pyruvate-ferredoxin oxidoreductase may be the source of reduced ferredoxin. However, this is speculative based on limited data. *Bacteroides* genetics are relatively straight forward, and it would be simple to test this hypothesis by generating pyruvate-ferredoxin oxidoreductase knockouts. It would also be useful to generate hydrogenase knockouts (though I do find the authors' conclusion about the responsible genes to be more compelling)."**
6. **Line 262: The conclusion in the section title and repeated in the section that "group B [FeFe]-hydrogenases to reoxidize ferredoxin" is not demonstrated by the data. The authors should either provide more data to support this conclusion or revise this section and acknowledge that the electron donor is unknown. One way to address this hypothesis would be to test a pyruvate-ferredoxin oxidoreductase KO strain for H₂ production.**

We agree with the reviewer's point about the speculative nature of our claims regarding the involvement of the pyruvate-ferredoxin oxidoreductase (PFOR) and reduced ferredoxin in *Bacteroides* H₂ production by the group B [FeFe]-hydrogenase. We have not been able to achieve comprehensive mutagenesis experiments such as those suggested by the reviewer within the timeframe of the revisions. Additionally, incorporating these experiments would have significantly expanded the already substantial complexity of the manuscript.

However, to address this concern and strengthen our findings, we have opted to instead take a biochemical approach. Specifically, we stimulated PFOR activity in *Bacteroides fragilis* cell lysate and measured the resulting H₂ production. After lysing the cells and stimulating the PFOR with the addition of pyruvate and coenzyme A (CoA), we observed a significant increase in H₂ production compared to the cell-extract-only control and buffer (Figure R4, in-text Fig. 4d). Additionally, inhibition of the PFOR with Nitazoxanide resulted in much lower H₂ production compared to the stimulation condition. Together these results demonstrate that H₂ production by *B. fragilis* is dependent on active PFOR and reduced ferredoxin as the electron donor to the group B [FeFe]-hydrogenase. We have included these results from lines 362-370.

We believe this biochemical approach provides valuable insight into the mechanism of H₂ production by the group B [FeFe]-hydrogenase and strengthens the conclusions of our study. These findings are also supported by the inferences from structural modelling, transcriptomics, and analyses of previous TraDIS datasets.

Figure R4. H₂ production (expressed as log₁₀ H₂ ppm) by *Bacteroides fragilis* cell extract (CE) upon stimulation of the pyruvate ferredoxin oxidoreductase with CoA and pyruvate (purple) in comparison to a CE-only control (light blue), buffer control (dark blue) and upon inhibition of the PFOR via the addition of nitazoxanide (light pink).

Other comments:

- 7. Bacteroides have previously been shown to produce H₂ during fermentation. Citing papers that have shown this would be appropriate and could strength some of arguments, in my opinion.**

We agree and have included citations to papers demonstrating Bacteroides H₂ production from fermentation on line 259¹⁻⁴

- 8. Title: I do not think the findings support the conclusion in the title that widespread hydrogenase “drives” fermentative growth. The authors show that hydrogenase activity is associated with fermentative growth but “drives” implies a far greater degree of essentiality than has been shown (or is likely in my opinion).**

We have changed the title to “A widespread hydrogenase supports fermentative growth of gut bacteria in health people”

- 9. Abstract: The statement “we show that metabolically flexible respiratory bacteria are the most abundant H₂ oxidizers in the gut, not sulfate reducers, methanogens, and acetogens as previously thought” is a bit misleading, in my opinion. Really the contention has been that “sulfate reducers, methanogens, and acetogens” are dominant H₂ oxidizers (irrespective of their level of abundance) and the new analyses don’t contradict that conclusion. The microbes identified by the authors as having flexible respiratory metabolisms use plenty of other oxidants than H₂ and so their higher abundance doesn’t imply that they are more important oxidizers.”**

We agree that the original statement in the abstract may have been too strongly worded. Our intent was to highlight that metabolically flexible bacteria, many of which encode hydrogenases, are more abundant in our datasets than the classical H₂ consuming groups, and that these organisms may represent a potentially significant but underexplored component of gut H₂ metabolism. However, we acknowledge that abundance alone does not demonstrate active H₂ oxidation, particularly given the metabolic flexibility of these organisms. To address this, we have revised the sentence on lines 50-52 to emphasise the potential rather than confirmed activity to: “Furthermore, we show that metabolically flexible respiratory bacteria may be dominant hydrogenotrophs in the gut, other than the previously described acetogens, methanogens, and sulfate reducers.”

- 10. Line 172: I am not sure but, at least in E. coli, the periplasmic nitrate reductase (NapA) is not believed to be a respiratory enzyme and so I am not sure it makes sense to include it in the analysis.**

This is a good point and we have amended to show the values for the respiratory nitrate reductase Nar instead. In Enterobacteriaceae, indeed the NarGHI complex is the main respiratory nitrate reductase, as it contributes directly to the proton motive force, whereas the periplasmic NapA does not. However, both enzymes can receive electrons from H₂ and thus still contribute to electron flow.

- 1 Chassard, C., Delmas, E., Lawson, P. A. & Bernalier-Donadille, A. Bacteroides xylanisolvens sp. nov., a xylan-degrading bacterium isolated from human faeces. *International journal of systematic and evolutionary microbiology* **58**, 1008-1013 (2008).
- 2 Harding, G., Sutter, V., Finegold, S. & Bricknell, K. Characterization of Bacteroides melaninogenicus. *Journal of clinical microbiology* **4**, 354-359 (1976).
- 3 Kazimierowicz, J., Dębowski, M. & Zieliński, M. Effectiveness of hydrogen production by Bacteroides vulgatus in psychrophilic fermentation of cattle slurry. *Clean Technologies* **4**, 806-814 (2022).
- 4 Suzuki, A. *et al.* Quantification of hydrogen production by intestinal bacteria that are specifically dysregulated in Parkinson's disease. *PLoS One* **13**, e0208313 (2018).

11. Line 190: It seems premature to conclude that “group B [FeFe]-hydrogenase appears to drive fermentative H₂ production” at this stage in the manuscript.

We have revised the sentence to read “Group B [FeFe]-hydrogenase likely contributes substantially to fermentative H₂ production” (now lines 200-202).

12. Line 231: I find the statement “Such *Clostridium* species appear to have evolved exceptionally rapid hydrogenases to enable vigorous growth in high nutrient conditions” unconvincing. These organisms express group A1 hydrogenases and this claim dismisses the observation that group A1 hydrogenases may be a more significant source of H₂. However, this possibility cannot be so easily dismissed. The “group B” expressing *Bacteroides* grow well too and are produce an order of magnitude less H₂ and the importance of group A1 hydrogenases cannot be that easily ruled out.

We thank the reviewer for pointing this out and have revised the sentence to improve clarity, as it was intended to be about the group A1 [FeFe]-hydrogenase as suggested by the reviewer, and not the group B. It now reads: “Such *Clostridium* species appear to have evolved these exceptionally rapid group A1 [FeFe]-hydrogenases to enable vigorous growth in high nutrient conditions such as experienced in this experimental setup. Though, the metagenomic and metatranscriptomic analysis suggests they are in low abundance in most stool and biopsy samples.” (lines 245-247)

13. Figure 3b: The authors should modify how this data is presented. There are so many zeros on the ppm axis that it’s challenging to compare the panels and recognize the substantial differences between strains in hydrogen production.

We appreciate and have carefully considered the reviewer’s suggestion and have trialled adjusting the axis scale accordingly. However, we believe that retaining the absolute values (in ppm), despite the presence of many zeros, best conveys the main message of this panel which is that substantial H₂ production occurs across a taxonomically diverse set of isolates, rather than providing a basis for direct comparison. Though, to facilitate direct comparison, we have altered the format of the figure to include the second heatmap as a separate panel (b) and thus make it a more distinct result (Fig. R5, in-text Fig. 3). This panel displays the average maximum H₂ production for each isolate. We hope this change improves the interpretability of this figure.

Figure R5. Hydrogenase transcription and activity across 18 bacterial gut isolates. (a) A heatmap showing the average transcription levels (expressed as \log_{10} transcripts per million, or TPM) of the catalytic subunit genes for hydrogenases and the terminal reductases associated with sulfidogenesis, succinogenesis, nitrate reduction, and aerobic respiration. **(b)** A heatmap showing the average maximum H_2 production for each isolate (expressed as \log_{10} ppm+1). In both heatmaps, results show means from biologically independent triplicates. *B. longum* and *C. mitsuokai* do not encode hydrogenases and so are used as negative controls. **(c)** Bacterial growth measured by optical density (OD_{600} , green lines) and H_2 production (% of headspace; pink lines), of representative isolates from chosen phyla over 24–72-hour periods ($n = 3$), where the lower detection threshold of the gas chromatograph is 1000 ppm, or 0.1% (dashed red line).

14. Line 344/Figure S8: Two questions/points about these analyses 1) Are the differences reported here statistically significant? The data and conclusions about the effect on gnotobiotic mouse colonization seem particularly unconvincing – as there is clearly mouse-to-mouse variability in both directions. 2) These rb-tn-seq experiments are done in complex communities within chambers that contain high concentrations of H₂. It is thus possible that the hydrogenases are functioning in the reverse direction within this context. This may be worth considering

We thank the reviewer for these comments, only a portion of the reported Tn mutant data is statistically significant, this error has now been rectified in Figure S11 and the main text, where only statistically significant findings are reported and discussed. These changes do not alter the overall discussion regarding the importance of the group B hydrogenase for *B. thetaiotaomicron* fitness in the presence of specific carbon sources or stress condition, but does undermine the conclusions surrounding gut colonisation, we have therefore changed this section of the text to read as follows (lines 390-406):

“To further contextualize the importance of the hydrogenases of *Bacteroides*, we interrogated previously published differential fitness datasets that leveraged random barcoded transposon sequencing (RB-TnSeq) of *B. thetaiotaomicron* mutant libraries exposed to a range of conditions¹. In this previous study, transposon mutants were obtained for both hydrogenases, suggesting they are not essential for growth. However, the group B [FeFe] hydrogenase was important for growth of *B. thetaiotaomicron* on fermentable carbon sources, especially glucose (**Fig. S11**). The hydrogenase was also important for adaptation on specific nitrogen sources such as L-valine (**Fig. S11**) and fitness under stress conditions in rich or defined media. The strongest fitness defect was during treatment with metronidazole, a broad-spectrum prodrug activated by reduced ferredoxin; this is consistent with the predicted role of the group B [FeFe]-hydrogenase in reoxidizing reduced ferredoxin, whereas mutants lacking the enzyme will accumulate in its reduced state and more quickly activate the prodrug. By contrast, transposon mutants interestingly showed increased fitness when exposed to sodium fluoride and fusidic acid. By contrast, the group A3 [FeFe] hydrogenase did not show statistically significant fitness defects under the conditions tested. Altogether, these analyses suggest the group B [FeFe]-hydrogenase enhances the fitness of *Bacteroides* across a range of conditions, though the bacterium can adapt to loss of the enzyme.”

1 Liu, H. *et al.* Functional genetics of human gut commensal *Bacteroides thetaiotaomicron* reveals metabolic requirements for growth across environments. *Cell reports* 34 (2021).

15. Line 417: The argument that electron acceptors such as fumarate, nitrate, and sulfoxides might be important is based solely on the observation that these genes are abundant, however, gene abundance does not necessarily predict activity. Also, other compounds can serve as electron acceptors. The authors should either scale back this claim or expand the discussion to include the observation that other, perhaps unidentified, electron acceptors may be important.

We agree with the reviewer that gene abundance of these reductase genes does not necessarily predict their activity, however, we direct the reviewer to Figure 1a in the manuscript that also describes their expression throughout gut metatranscriptomes, and demonstrates that while they are abundant, they are also transcribed. We have rephrased the sentence for clarity (see below). While expression, again, does not necessarily indicate activity, we believe this is a solid indication that these reductase genes are likely an important component of potential gut H₂ consumption-related metabolic pathways.

We also appreciate the reviewer highlighting that there are potentially other compounds that can serve as electron acceptors for H₂ consumption pathways, and we agree! We have incorporated this hypothesis by including the following (lines 465-473):

“Instead, it is nevertheless likely that respiratory bacteria that use electron acceptors such as fumarate, nitrate, and sulfoxides play a potentially dominant role in gut H₂ consumption, given the abundance and transcription of associated hydrogenases and respiratory reductases in both stool and biopsy metagenomes and stool metatranscriptomes, though activity measurements are required to validate this. Additionally, recent work has highlighted the potential for numerous additional organic electron acceptors in the gut, including dietary and host-derived compounds¹, that could also be potentially involved in H₂ oxidation. Enterobacteriaceae may be particularly important hydrogenotrophs.”

1 Little, A. S. *et al.* Dietary-and host-derived metabolites are used by diverse gut bacteria for anaerobic respiration. *Nature microbiology* 9, 55-69 (2024).

Reviewer #2

- 1. I have no major concerns except that the authors may be over-interpreting some of the data of the modelling. For example, “AlphaFold2 modelling (Fig. S3 & S4) confirms group B [FeFe]-hydrogenases are structurally conserved between Bacteroides species (Fig. S5) and form monomers with two distinct globular domains”. I recommend that this should be toned down or better supported by experimental data (for example that these enzymes are monomers).**

We agree with the reviewer that AlphaFold2 modelling does not confirm the structure of these hydrogenases, since it is a predictive tool. We thank the reviewer for catching this oversight. We have changed this sentence to say “AlphaFold2 modelling predicts” instead of “AlphaFold2 modelling confirms”. Our AlphaFold2-multimer modelling predicts that the group B [FeFe]-hydrogenase does not form a homodimer, homotrimer, or homotetramer. However, this doesn’t rule out other protein-protein interactions and formation of heterooligomers, therefore the reviewer is correct that we should not definitively state that group B [FeFe]-hydrogenases are monomers. We have removed the claim about its oligomeric state, and the sentence now reads as: “AlphaFold2 modelling (Fig. S3 & S4) predicts group B [FeFe]-hydrogenases are structurally conserved between Bacteroides species (Fig. S5) and contain two distinct globular domains” (line 283).

Reviewer #3

- 1. Page 5, lines 137-138: The by-sentence “though operating alongside other enzymes” appears trivial. The authors may have meant “though operating alongside other H₂-producing hydrogenases”, which would make sense.**

We thank the reviewer for picking this up and have adjusted the sentence as suggested to read: “Altogether, group B [FeFe]-hydrogenases potentially drive most H₂ production in the gut, though operate alongside other H₂ -producing hydrogenases” (line 141-142).

- 2. Page 5, lines 145-146: The authors identify group B [FeFe]-hydrogenase as presumably most abundant hydrogenase in the human gut and this hydrogenase type is widespread in Bacteroides. To contextualize the above findings, the relative (average) abundance of Bacteroides species in the gut microbiome should be mentioned here and/or elsewhere.**

We agree this information would be ideal to contextualise the overall importance of the group B [FeFe]-hydrogenase across the community. As such we have run MetaPhlan4 on the stool and biopsy metagenomes to identify the relative abundance of *Bacteroides* within this community. We have found that *Bacteroides* were among the most abundant genera detected in our metagenomes, detected in 97% of stool metagenome samples and 100% of biopsy

metagenome samples (**Table S3**). Within these samples *Bacteroides* were also detected to be highly abundant, comprising an average of 25% of the stool and 45% of the biopsy communities (**Table S3**), further supporting the importance of the group B [FeFe]-hydrogenase in the lifestyles of dominant fermentative gut taxa. This information is now incorporated into the manuscript on lines 153-158 and in supplementary table S3.

- 3. Page 6, line 170 (and many other places!):** The term “gene expression” encompasses both gene transcription and the translation of the corresponding mRNA into proteins. Thus, there are three levels (gene, transcript, protein) that are discussed in detail in this manuscript and should be named accordingly to avoid confusing the reader. In the example in line 170, the statement “enzymes except fumarate reductase” should be replaced by “genes except the genes encoding fumarate reductase” because transcriptome data are being discussed here. Another example of the need for more precise wording can be found on page 8, lines 238–240. Here, the genome of an actinobacterium carries a hydrogenase gene that is transcribed, but the transcript does not appear to be translated, i.e. no active enzyme is produced (or synthesized) because no H₂ has been observed. Therefore, it is inappropriate to claim that a group B [FeFe]-hydrogenase is expressed in this species, unless the authors define expression as mere gene transcription, which is currently not the case. This applies also for the titles of Figures 1-3. They should be “Abundance, expression and distribution of hydrogenase genes and H₂ -related metabolic genes throughout the human gut.”, “Phylogenomic tree showing distribution of hydrogenase genes among 812 bacterial isolates from the human gut.” and “Hydrogenase gene transcription and activity across 18 bacterial gut isolates.”, respectively. The authors should check the entire manuscript for correct wording in this regard.”

AND

- 4. Page 8, line 246:** Another example for incorrect wording. “For all seven strains, the group B [FeFe]-hydrogenase genes were transcribed at high levels” (which indicates hydrogenase gene expression since H₂ was observed in the headspace of the cultures).

We are grateful for the reviewer’s thorough observations, which has prompted us to refine our terminology and improve clarity throughout the manuscript. We have carefully reviewed the manuscript and revised multiple instances within the results section, and figure legends to more accurately distinguish between genes, transcripts, and proteins/enzymes, and to ensure precise use of terms such as “expression”.

- 5. Page 7, line 190:** “..., much of which is likely to be partially recycled by respiratory...” (as mentioned in the introduction, H₂ gas is expelled as flatus or exhaled).

We thank the reviewer for this comment, while we do mention in the introduction that a lot of H₂ is excreted as breath or flatus, we suggest that most is recycled by other H₂ consuming organisms for these processes (lines 58-60) which is supported by the abundance and transcription of H₂ consuming hydrogenase genes and also known and potentially associated anaerobic respiratory reductase genes (Figure 1a, Figure 2 & Figure 3a).

- 6. Page 7, line 212:** It is not clear to the reader what is meant by “during fermentative survival”. If group 4a [NiFe]-hydrogenases are thought to be required for fermentative growth of these bacteria, this should be clearly stated.

The authors apologise for the lack of clarity surrounding this phrase and appreciate the reviewer bringing it to our attention. Here, we use the phrase “fermentative survival” as a proxy for bacterial persistence or stationary phase. Essentially, we are aiming to highlight that the

group 4a [NiFe]-hydrogenase encoded and expressed by *Collinsella aerofaciens* in Figure 3c and *Necropsobacter rosorum* in Figure S1 is primarily thought to be most actively producing H₂ during persistence in response to nutrient limitation faced in these experimental conditions, in comparison to the group B [FeFe]-hydrogenase which is most active during fermentative growth. We have altered the sentence to read:

“...and H₂ was produced during stationary phase in bacteria with genes for the group 4a [NiFe]-hydrogenase-containing formate hydrogenylases (*Collinsella aerofaciens*, *Necropsobacter rosorum*), in line with their confirmed roles” (lines 224-227)

7. Page 9, line 279: The designation “His-ligated [4Fe-4S] cluster” is misleading, as it is coordinated by three cysteines and one histidine.

The term “His-ligated [4Fe-4S] cluster” is common in hydrogenase literature and in descriptions of Cpl¹⁻³. It refers to a [4Fe-4S] cluster that is coordinated by one histidine and three cysteines. However, not all readers may be aware of this and the term may mislead readers to think that four histidines coordinate the [4Fe-4S] cluster. Therefore, we have changed this line to be more descriptive: “whereas the group A enzyme contains a [2Fe2S] ferredoxin-like domain and a 3-Cys, 1-His ligated [4Fe4S] cluster...”. (lines 295-297)

- 1 Mulder, D. W. *et al.* Insights into [FeFe]-hydrogenase structure, mechanism, and maturation. *Structure* **19**, 1038-1052 (2011).
- 2 Rodríguez-Maciá, P. *et al.* His-ligation to the [4Fe-4S] subcluster tunes the catalytic bias of [FeFe] hydrogenase. *Journal of the American Chemical Society* **141**, 472-481 (2018).
- 3 Bak, D. W. & Elliott, S. J. Alternative FeS cluster ligands: tuning redox potentials and chemistry. *Current opinion in chemical biology* **19**, 50-58 (2014).

8. Page 9, line 279: “This ferredoxin-like domain is unusual in that its iron-sulfur clusters are modelled to be distant from the main body”. The structures presented in this study are based solely on AlphaFold2 calculations. Although the authors also used AF-Cluster to determine possible conformational changes, the proposed structures might be wrong. It is still only an assumption that the ferredoxin domain is beyond electron transfer distance to the corresponding [4Fe-4S]-cluster. It is important to mention the unusual conformation of the ferredoxin-like domain but is not a hard fact. The authors could easily include ferredoxin in their Alphafold2 calculations to support their assumption that the ferredoxin interaction enables productive electron transfer.”

While the reviewer is correct that the approximate 22 Å distance between two iron-sulfur clusters could still support electron transfer, we decided to make the conservative claim that they may not support electron transfer, since inter-cluster distances greater than 14 Å are generally not physiologically productive¹. We can change this section to not rule out the possibility of electron transfer and stick with established facts:

“This ferredoxin-like domain is unusual in that its iron-sulfur clusters are distant from the main body of the enzyme as they are separated from the nearest [4Fe4S] cluster in the H-cluster domain by an edge-to-edge distance of at least 22 Å (Fig. 4a). Even after accounting for conformational flexibility of the interdomain loops using AF-Cluster, the greater than 14 Å distance between these two clusters would likely limit the rate of effective electron transfer.”

We respectfully disagree with the reviewer that “the authors could easily include ferredoxin in their Alphafold2 calculations to support their assumption that the ferredoxin interaction enables productive electron transfer.” This would require running AlphaFold2-multimer between the group B [FeFe]-hydrogenase and all ferredoxin and all proteins containing ferredoxin domains in the *Bacteroides* genome with a protocol like AlphaPullDown, and this is computationally expensive. We also question the applicability of AlphaFold2 for this specific question. If the group B [FeFe]-hydrogenases cognate electron donating partner makes a weak and transient interaction (as ferredoxin interactions generally are), then AlphaFold2 may not

be able to predict it. AlphaFold2 is suitable for predicting protein-protein interactions within stable long-lived complexes, since they contain co-evolving residues. However, weak transient interactions driven by electrostatic attraction may lack the deep co-evolutionary information needed for AlphaFold2 and may lead to false negative predictions.

9. Page 9, line 287: “Trimeric enzymes” is a bit vague. Group A3 [FeFe]-hydrogenases usually form dimers of heterotrimers.

This is correct and we thank the reviewer for catching this oversight. We have updated the sentence to read: “Structural predictions also indicated that the group A3 [FeFe]-hydrogenases of *Bacteroides* are a homodimer of heterotrimers...” (line 303-305). We have also removed other instances of describing the group A3 [FeFe]-hydrogenases as “trimeric” throughout this study.

10. Page 9, lines 293-294: The authors demonstrate hydrogenase activity in cell lysates using dithionite-reduced methyl viologen as electron donor. The title of the chapter reads “Bacteroides use group B [FeFe]-hydrogenases to reoxidize ferredoxin during fermentation”. Although it is highly likely that ferredoxin is the native electron donor based on the knowledge accumulated in the literature and especially in this study, there is still no direct evidence. This should be taken into account.

We absolutely agree. As mentioned in the responses to reviewer #1’s comments 5 & 6 above, we have now addressed this oversight by demonstrating that stimulating the activity of the pyruvate-ferredoxin oxidoreductase, which reduces ferredoxin, results in enhanced H₂ production of *Bacteroides fragilis* cell extract. We believe this is further evidence pointing towards the group B [FeFe]-hydrogenase reliance on reduced ferredoxin as an electron donor.

11. Figure 4: This figure should only contain fields a-c. Panel d (EPR spectra) can be moved to the supplement and panel e should be an independent figure. Here, a seven-compartment box with the seven strains used for the transcriptional analysis should be presented in the figure. The legend (page 26) should explain the abbreviations of the different enzymes presented in the figure. Gray panels, as mentioned in the legend, are not present in the figure.

We appreciate the reviewer’s feedback regarding the layout and clarity of Figure 4. While we agree that some adjustments are necessary, we respectfully disagree with the suggestion to include an additional seven panels to display the same metabolic figure separately for each *Bacteroides* strain. We feel that this information would become redundant due to the general similarities between the transcription levels of most genes within the pathway and increase complexity of the figure without adding in too much extra interpretative value. Instead, we have improved clarity by adding the species initials above each heatmap box (See Fig R6).

In regard to the reviewer’s other suggestions to improve figure 4, we have moved the EPR spectra (previously panel d) to the supplementary (Figure S8) clarified all abbreviations in the figure legend and removed the reference to the grey boxes. We hope the changes sufficiently address the concerns raised and improve the figure’s overall clarity.

(f) Summary of the expression levels of fermentation genes in seven enteric *Bacteroides* isolates, including the group B and group A3 [FeFe]-hydrogenases. Expression is shown as TPM in boxes in the order of *B. caccae* (*Bc*), *B. faecis* (*Bfa*), *B. fragilis* (*Bfr*), *B. thetaiotaomicron* (*Bt*), *B. dorei* (*Bd*), *B. plebius* (*Bp*), and *B. vulgatus* (*Bv*) under each relevant gene. Enzyme abbreviations are as follows: NqrABCDEF = Na⁺-translocating NADH:quinone oxidoreductase, Mdh = malate dehydrogenase, Ndh1 (Nuo) = NADH:ubiquinone oxidoreductase (Complex I), PckA = phosphoenolpyruvate carboxykinase, Pyk = pyruvate kinase, MaeB = NADP-dependent malic enzyme, RnfABCDEG = ferredoxin:NAD⁺ oxidoreductase, Ndh2 = type II

NADH dehydrogenase, FumA = fumarase, PflB = pyruvate formate-lyase, Por = pyruvate:ferredoxin oxidoreductase, NifJ = pyruvate:flavodoxin oxidoreductase, FrdAB-CytB = fumarate reductase, Pta = phosphotransacetylase, Ack = acetate kinase, MmdABCG = methylmalonyl CoA decarboxylase, ScpC = propionyl-CoA/Succinate-CoA transferase, SucCD = succinyl-CoA synthetase, Mce = metabolite transporter, MutAB = methylmalonyl-CoA mutase.”

Figure 4. Metabolic integration, predicted structure, and biochemical activity of the group B [FeFe]-hydrogenases from *Bacteroides*. (a) A superposition of representative structures of the group A [FeFe]-hydrogenase (*Clostridium pasteurianum* Cpl; X-ray crystallography; PDB: 6GM2⁷¹) and the group B [FeFe]-hydrogenase (*Bacteroides fragilis*; AlphaFold2). Portions where the two [FeFe]-hydrogenase groups show structural similarity (where RMSD <2 Å) are highlighted in bold and black outline. Divergence between the two structures (RMSD >2 Å) is depicted as transparent with no outline. RMSD: root mean square deviation. (b) Top-ranked predicted protein structure (AlphaFold2) of the *B. fragilis* group B [FeFe]-hydrogenase with putative [FeFe]-hydrogenase cofactors modelled. The predicted H-cluster site is shown in focus with conserved residues coordinating with the H-cluster labelled in green. Four iron-sulfur clusters are predicted to coordinate with conserved cysteines throughout the protein, labelled A1 to A4. (c) H₂ production (measured by GC) monitored from cell lysates activated by addition of [2Fe]^{adt}. Results are shown for the group B [FeFe]-hydrogenases of *B. fragilis*, *B. vulgatus*, and *B. thetaiotaomicron*, as well as the group A3 [FeFe]-hydrogenase of *B. thetaiotaomicron*. Activities were normalised for number of cells used (nmol H₂ min⁻¹ OD₆₀₀⁻¹) and error bars reflect standard deviation from biological triplicates. All enzymes were expressed in *E. coli* BL21(DE3) cells. The strain expressing the prototypical group A1 [FeFe]-hydrogenase from *Chlamydomonas reinhardtii* (CrHydA1) was used as a positive control, while “Blank” represents the same strain but containing an empty vector that was also added with [2Fe]^{adt}. (d) Summary of the expression levels of fermentation genes in seven enteric *Bacteroides* isolates, including the group B and group A3 [FeFe]-hydrogenases. Expression is shown as TPM in boxes in the order of *B. caccae* (Bc), *B. faecis* (Bfa), *B. fragilis* (Bfr), *B. thetaiotaomicron* (Bt), *B. dorei* (Bd), *B. plebius* (Bp), and *B. vulgatus* (Bv) under each relevant gene. Enzyme abbreviations are as follows: NqrABCDEF = Na⁺-translocating NADH:quinone oxidoreductase, Mdh = malate dehydrogenase, Ndh1 (Nuo) = NADH:ubiquinone oxidoreductase (Complex I), PckA = phosphoenolpyruvate carboxykinase, Pyk = pyruvate kinase, MaeB = NADP-dependent malic enzyme, RnfABCDEG = ferredoxin:NAD⁺ oxidoreductase, Ndh2 = type II NADH dehydrogenase, FumA = fumarase, PflB = pyruvate formate-lyase, Por = pyruvate:ferredoxin oxidoreductase, NifJ = pyruvate:flavodoxin oxidoreductase, FrdAB-CytB = fumarate reductase, Pta = phosphotransacetylase, Ack = acetate kinase, MmdABCG = methylmalonyl CoA decarboxylase, ScpC = propionyl-CoA/Succinate-CoA transferase, SucCD = succinyl-CoA synthetase, Mce = metabolite transporter, MutAB = methylmalonyl-CoA mutase. (e) *B. fragilis* H₂ production (ppm) relative to growth (OD₆₀₀) in the presence (pink line) and absence (green line) of hemin. (f) H₂ production (ppm) by *B. fragilis* cell extract upon stimulation of the pyruvate ferredoxin oxidoreductase with CoA and pyruvate (purple) in comparison to a cell-extract-only control (pink) and buffer (grey).

12. Page 11, lines 344-360: It should be better clarified that the barcoded transposon mutant library of *B. thetaiotaomiron* was already available. The observations that, e.g., transposon mutants were obtained for both hydrogenases currently sound as they were made by the authors of this manuscript. This also applies to the colonization experiments of germ-free mice (Figure S8F should read S8E). The negative effect of metronidazole on the fitness of group B [FeFe] hydrogenase gene mutants is quite clear and can be nicely explained. The same mutants appear to have a growth advantage in the presence of fusidic acid and NaF. The authors could comment on this as well.

We thank the reviewer for these comments and agree that we could improve the clarity of this section of the manuscript. We have modified the text to ensure it is clear that the Tn mutant library was previously constructed by another group, and we have reanalysed these datasets with a focus on each *B. thetaiotaomicron* hydrogenase. We have also mentioned the fitness advantages reported for exposure to fusidic acid and sodium fluoride, though cannot currently speculate on the basis of this. The modifications are as follows (Fig. R7, in-text Fig. S11) and lines 390-406.

“To further contextualize the importance of the hydrogenases of *Bacteroides*, we interrogated previously published differential fitness datasets that leveraged random barcoded transposon sequencing (RB-TnSeq) of *B. thetaiotaomicron* mutant libraries exposed to a range of conditions⁹⁶. In this previous study, transposon mutants were obtained for both hydrogenases, suggesting they are not essential for growth. However, the group B [FeFe] hydrogenase was important for growth of *B. thetaiotaomicron* on fermentable carbon sources, especially glucose (**Fig. S11**). The hydrogenase was also important for adaptation on specific nitrogen sources such as L-valine (**Fig. S11**) and fitness under stress conditions in rich or defined media. The strongest fitness defect was during treatment with metronidazole, a broad-spectrum prodrug activated by reduced ferredoxin; this is consistent with the predicted role of the group B [FeFe]-hydrogenase in reoxidizing reduced ferredoxin, whereas mutants lacking the enzyme will accumulate in its reduced state and more quickly activate the prodrug. By contrast, transposon mutants interestingly showed increased fitness when exposed to sodium fluoride and fusidic acid. By contrast, the group A3 [FeFe] hydrogenase did not show statistically significant fitness defects under the conditions tested. Altogether, these analyses suggest the group B [FeFe]-hydrogenase enhances the fitness of *Bacteroides* across a range of conditions, though the bacterium can adapt to loss of the enzyme.”

Figure R7. Differential fitness data of *B. thetaiotaomicron* random barcoded transposon mutant libraries exposed to various conditions. Data adapted from (1) (Fitness Browser <https://fit.genomics.lbl.gov/cgi-bin/genesFit.cgi?orgId=Btheta&locusId=351362&locusId=349652&around=0>). Statistically significant Group B [FeFe] hydrogenase transposon mutant fitness in different experimental conditions (different carbon sources, nitrogen sources, upon exposure to various stress conditions in rich brain heart infusion media (BHIS) or defined Varel and Bryant media (VBM)). Fitness values are represented as log₂ ratios that indicate the change in abundance of mutants under each condition. A fitness score of 0 indicates mutants with an insertion in this gene did not confer a fitness defect, fitness score of < 0 means the gene was important for fitness under this condition, fitness of > 0 means that the loss of this gene gave mutants a growth advantage. Fitness score of ±1 is indicated on each graph as this score indicates that these mutants were 50% more or less abundant after exposure. Fitness scores between ±1 indicate subtle phenotypes, whereas fitness scores less than -2 or greater than 2 are considered strong phenotypes. All results shown are statistically significant with a reported absolute t-score of ≥4, a threshold adopted by the original authors².

- 1 Page, C. C., Moser, C. C., Chen, X. & Dutton, P. L. Natural engineering principles of electron tunnelling in biological oxidation–reduction. *Nature* **402**, 47-52 (1999).
- 2 Liu, H. *et al.* Functional genetics of human gut commensal *Bacteroides thetaiotaomicron* reveals metabolic requirements for growth across environments. *Cell reports* **34** (2021).

13. Tables S1, S2, S4 and S7 (currently provided as Excel files with cryptic names) should be given unique names that should include S1, S2, S4 and S7. Furthermore, their content could be made more accessible with extended table legends.

The authors thank the reviewer for identifying this, and we have addressed this concern by giving each table a specific name and accompanying table legend as follows.

Table S1. “*Table S1 - Hydrogenase abundance and expression in gut metagenomes and metatranscriptomes.xlsx*”. A comprehensive analysis of the abundance, transcription and taxonomic origin of genes encoding hydrogenases and H₂-cycling-related metabolic genes throughout stool metagenome and metatranscriptome samples and biopsy metagenome samples (with accompanying ENA run accession numbers),. The data is organised as per the following tabs:

1. **“Stool MetaG Hyd-sbg cpg”**

- Abundance values, expressed as copies per genome (cpg), of genes encoding hydrogenase subgroups throughout the *stool metagenome* samples.

2. **“Stool MetaG Hyd-sbg RPKM”**

- Abundance values, expressed as reads per kilobase million (RPKM), of genes encoding hydrogenase subgroups throughout the *stool metagenome* samples.

3. **“Stool MetaG Hyd-mg Analysis”**

- Presence/absence analysis and average abundance values (cpg and RPKM) of groupings of the genes encoding the main hydrogenase groups ([NiFe]- and [FeFe]-hydrogenase) throughout the *stool metagenome* samples.

4. **“Biopsy MetaG Hyd-sbg cpg”**

- Abundance values, expressed as copies per genome (cpg), of genes encoding hydrogenase subgroups throughout the *biopsy metagenome* samples.

5. **“Biopsy MetaG Hyd-sbg RPKM”**

- Abundance values, expressed as reads per kilobase million (RPKM), of genes encoding hydrogenase subgroups throughout the *biopsy metagenome* samples.

6. **“Biopsy MetaG Hyd-mg Analysis”**

- Presence/absence analysis and average abundance values (cpg and RPKM) of groupings of the genes encoding the main hydrogenase

groups ([NiFe]- and [FeFe]-hydrogenase) throughout the *biopsy metagenome* samples.

7. **“Stool MetaT Hyd-sbg RPKM”**

- Transcription values, expressed as reads per kilobase million (RPKM), of genes encoding hydrogenase subgroups throughout the *stool metatranscriptome* samples.

8. **“Stool MetaT Hyd-mg Analysis”**

- Transcribed/Not transcribed analysis and average transcription values (RPKM) of genes encoding the main hydrogenase groupings ([NiFe]- and [FeFe]-hydrogenase) throughout the *stool metatranscriptome* samples.

9. **“Stool MetaT FuncGenes Analysis”**

- Transcribed/not transcribed analysis and transcription values (RPKM), of H₂-cycling-related metabolic genes throughout the *stool metatranscriptome* samples.

10. **“Stool & Biopsy MetaG FuncGenes”**

- Presence/absence analysis and abundance values (cpg & RPKM), of H₂-cycling-related metabolic genes throughout the *stool* (yellow) & *biopsy* (orange) *metagenome* samples.

11. **“Stool v Biopsy MetaG Av. cpg”**

- Comparative analysis of the average abundance (cpg) of genes encoding hydrogenase subgroups between *stool* vs *biopsy* metagenomes

12. **“Transcr-Abun FuncGenes RPKM”**

- Transcription and abundance values (RPKM), and corresponding ratios, of H₂-cycling-related metabolic genes across paired *stool metagenome* and *metatranscriptome* samples.

13. **“Transcr-Abun Hyd-sbg RPKM”**

- Transcription and abundance values (RPKM), and corresponding ratios, of genes encoding hydrogenase subgroups across paired *stool metagenome* and *metatranscriptome* samples.

14. **“Transcr-Abun Hyd-mg RPKM”**

- Transcription and abundance values (RPKM), and corresponding ratios, of groupings of genes encoding the main hydrogenase subgroups across paired *stool metagenome* and *metatranscriptome* samples.

15. **“Genus Hyd-sbg sum(RPKM)”**

- The genera with the highest abundance from *stool* and *biopsy* metagenomes, and transcriptional activity from *stool metatranscriptomes*

(expressed as sum(RPKM)), of genes encoding hydrogenase subgroups as inferred from read-mapping.

16. **“Phylum abund. MetaG sum(RPKM)”**

- Phylum-level abundance (sum(RPKM)) of genes encoding hydrogenase subgroups throughout the *stool* and *biopsy metagenome* samples.

Table S2. “Table S2 - Taxonomy and H₂ cycling related metabolic capacity of AusMICC gut isolates.xlsx”. Taxonomic classifications for 812 human gut isolate genomes from the Australian Microbiome Culture Collection (AusMicc) (with accompanying ENA run accession numbers), along with a summary of the presence or absence of genes encoding hydrogenase subgroups and other H₂ cycling-related metabolic genes across these genomes. The data is organised in the following tabs:

1. **“Tax & Funcgene Hit Summary”**

- A summary of the phyla and genus level taxonomic identification of each isolate genome, along with hits to hydrogenase subgroup and other H₂ cycling related metabolic genes, along with the database information for each hit.

2. **“Gene Count and % Analysis”**

- Count and percentage presence of the genes encoding hydrogenase subgroups and other H₂ cycling related metabolic genes across all isolate genomes.

3. **“Phylum and Genus Analysis”**

- The dominant phyla and genera (identified by GTDB-Tk) that encode hydrogenase subgroup and other H₂ cycling related metabolic genes across the isolate genome collection.

4. **“GTDB-Tk IDs”**

- GTDB-Tk-based taxonomic classification of each isolate genome.

5. **“CheckM”**

- Quality assessment metrics obtained from CheckM for each isolate genome.

6. **“Bacteroides Hydrogenases”**

- Number of overall hydrogenase gene hits detected to be encoded by each *Bacteroides* isolate genome.

Table S3. “Table S3 - Taxonomic classification of stool and biopsy metagenomes.xlsx”. The relative abundance (%) of microbial taxa in the stool and biopsy metagenome samples, as profiled by MetaPhlan4. Analysis of the proportion of metagenome samples that detect *Bacteroides*, and average relative abundance of *Bacteroides* within those samples is also included. The data is organised via the following tabs:

1. **“Biopsy metagenome taxonomy”**.
 - The relative abundance (%) of microbial taxa detected within the *biopsy metagenome* samples. Rows represent the taxonomic clades from kingdom to species level, and columns represent individual metagenome samples. The relative abundance of *Bacteroides* across samples is highlighted in red, and analysis of abundance and prevalence of *Bacteroides* is included.
2. **“Biopsy MetaG ENA Accessions”**
 - A summary of the *biopsy metagenome* sample ID’s used in the analysis, the corresponding filenames and their ENA run accession numbers that can be used to access the files.
3. **“Stool metagenome taxonomy”**.
 - The relative abundance (%) of microbial taxa detected within the *stool metagenome* samples. Rows represent the taxonomic clades from kingdom to species level, and columns represent individual metagenome samples. The relative abundance of *Bacteroides* across samples is highlighted in red, and analysis of abundance and prevalence of *Bacteroides* is included.

Table S5. “Table S5 - AusMiCC Gut Isolate transcriptome summaries.xlsx”. Summaries of the transcriptomic analysis of all 18 gut isolates obtained from AusMiCC, including transcripts per million values for all transcribed and annotated genes, and accompanying KEGG ID’s and modules. The data is organised into the following tabs:

1. **“Isolate Genome IDs & Accessions”**
 - A list of the AusMiCC isolate IDs and the corresponding species identification, ENA run accession numbers for the isolate genomes, and ENA run accession numbers for the transcriptome (RNA-seq file) of each isolate replicate (n=3).
2. **“B. caccae”**

- A list of all transcribed and annotated genes and their quantification (transcripts per million, TPM) from triplicate cultures of *Bacteroides caccae*, including KEGG ID and module information, and the DIAMOND homology-based identification of the hydrogenase and related H₂ cycling genes.
3. **“B. dorei”**
 - A list of all transcribed and annotated genes and their quantification (transcripts per million, TPM) from triplicate cultures of *Bacteroides dorei*, including KEGG ID and module information, and the DIAMOND homology-based identification of the hydrogenase and related H₂ cycling genes.
 4. **“B. faecis”**
 - A list of all transcribed and annotated genes and their quantification (transcripts per million, TPM) from triplicate cultures of *Bacteroides faecis*, including KEGG ID and module information, and the DIAMOND homology-based identification of the hydrogenase and related H₂ cycling genes.
 5. **“B. fragilis”**
 - A list of all transcribed and annotated genes and their quantification (transcripts per million, TPM) from triplicate cultures of *Bacteroides fragilis*, including KEGG ID and module information, and the DIAMOND homology-based identification of the hydrogenase and related H₂ cycling genes.
 6. **“B. plebius TM”**
 - A list of all transcribed and annotated genes and their quantification (transcripts per million, TPM) from triplicate cultures of *Bacteroides plebius*, including KEGG ID and module information, and the DIAMOND homology-based identification of the hydrogenase and related H₂ cycling genes.
 7. **“B. theta”**
 - A list of all transcribed and annotated genes and their quantification (transcripts per million, TPM) from triplicate cultures of *Bacteroides thetaiotaomicron*, including KEGG ID and module information, and the DIAMOND homology-based identification of the hydrogenase and related H₂ cycling genes.
 8. **“B. vulgatus”**

- A list of all transcribed and annotated genes and their quantification (transcripts per million, TPM) from triplicate cultures of *Bacteroides vulgatus*, including KEGG ID and module information, and the DIAMOND homology-based identification of the hydrogenase and related H₂ cycling genes.

9. **“C. baratii”**

- A list of all transcribed and annotated genes and their quantification (transcripts per million, TPM) from triplicate cultures of *Clostridium baratii*, including KEGG ID and module information, and the DIAMOND homology-based identification of the hydrogenase and related H₂ cycling genes.

10. **“C. perfringens”**

- A list of all transcribed and annotated genes and their quantification (transcripts per million, TPM) from triplicate cultures of *Clostridium perfringens*, including KEGG ID and module information, and the DIAMOND homology-based identification of the hydrogenase and related H₂ cycling genes.

11. **“A. hadrus”**

- A list of all transcribed and annotated genes and their quantification (transcripts per million, TPM) from triplicate cultures of *Anaerostipes hadrus*, including KEGG ID and module information, and the DIAMOND homology-based identification of the hydrogenase and related H₂ cycling genes.

12. **“D. longicatena”**

- A list of all transcribed and annotated genes and their quantification (transcripts per million, TPM) from triplicate cultures of *Dorea longicatena*, including KEGG ID and module information, and the DIAMOND homology-based identification of the hydrogenase and related H₂ cycling genes.

13. **“G. formicilis”**

- A list of all transcribed and annotated genes and their quantification (transcripts per million, TPM) from triplicate cultures of *Gemmiger formicilis*, including KEGG ID and module information, and the DIAMOND homology-based identification of the hydrogenase and related H₂ cycling genes.

14. **“N. rosorum”**

- A list of all transcribed and annotated genes and their quantification (transcripts per million, TPM) from triplicate cultures of *Necropsobacter rosorum*, including KEGG ID and module information, and the DIAMOND homology-based identification of the hydrogenase and related H₂ cycling genes.

15. **“C. aerofaciens”**

- A list of all transcribed and annotated genes and their quantification (transcripts per million, TPM) from triplicate cultures of *Collinsella aerofaciens*, including KEGG ID and module information, and the DIAMOND homology-based identification of the hydrogenase and related H₂ cycling genes.

16. **“O. umbonata”**

- A list of all transcribed and annotated genes and their quantification (transcripts per million, TPM) from triplicate cultures of *Olsenella umbonata*, including KEGG ID and module information, and the DIAMOND homology-based identification of the hydrogenase and related H₂ cycling genes.

17. **“F. varium”**

- A list of all transcribed and annotated genes and their quantification (transcripts per million, TPM) from triplicate cultures of *Fusobacterium varium*, including KEGG ID and module information, and the DIAMOND homology-based identification of the hydrogenase and related H₂ cycling genes.

18. **“C. mitsuokai”**

- A list of all transcribed and annotated genes and their quantification (transcripts per million, TPM) from triplicate cultures of *Catenibacterium mitsuokai*, including KEGG ID and module information, and the DIAMOND homology-based identification of the hydrogenase and related H₂ cycling genes.

19. **“B. longum”**

- A list of all transcribed and annotated genes and their quantification (transcripts per million, TPM) from triplicate cultures of *Bifidobacterium longum*, including KEGG ID and module information, and the DIAMOND homology-based identification of the hydrogenase and related H₂ cycling genes.

20. **“TPM Summary - All”**

- The combined raw transcriptomic information expressed as TPM (outputted by Salmon) for each isolate.
21. **“Bacteroides Hyd TPM Summary”**
 - A quantified transcription summary of the hydrogenase genes encoded by *Bacteroides* isolates.
 22. **“Av. Phyla TPM - DIAMOND genes”**
 - Average transcript abundance (TPM) of hydrogenase subgroup and other H₂ cycling related metabolic genes across all encoding isolates within each phylum.
 23. **“Isolate genome DIAMOND hits”**
 - A list of all hits to each isolate genome identified through homology-based searches using DIAMOND against an in-house database containing hydrogenase and other H₂ cycling related metabolic protein sequences.
 24. **“Bacteroides KEGG ID”**
 - Transcript abundance data (TPM) for the KEGG-annotated genes across *Bacteroides* isolate genomes.

Table S8. “Table S8 - Hydrogenase gene abundance in stool metagenomes of healthy vs unhealthy individuals”. A summary of the relative abundances (in copies per genome, cpg) of hydrogenase subgroup genes and other H₂-cycling related metabolic genes across stool metagenomes of healthy individuals compared to those experiencing various chronic disease states. The data is organised in the following tabs:

1. **“FuncGenes abund.”**
 - Relative abundances (cpg) of H₂-cycling related functional/metabolic genes for each stool metagenome sample, along with their health status (healthy or various chronic disease states).
2. **“FuncGenes abund. across disease”**
 - A summary of the average relative abundance (mean cpg + standard deviation) of hydrogenase main groups and various H₂ cycling related metabolic genes for the healthy state, and for each disease state.
3. **“Hyd-sbgr abund.”**
 - Relative abundances (cpg) of hydrogenase subgroup genes for each stool metagenome sample, along with their health status (healthy or various chronic disease states).
4. **“Hyd-sbgr abund. across diseases”**

- A summary of the average relative abundance (mean cpg + standard deviation) of hydrogenase subgroup genes for the healthy state, and for each disease state.

Responses to Reviewer #1 (Remarks to the Author):

The authors have made substantial efforts to address the concerns raised in the initial review, and the revised manuscript shows significant improvement through additional experimental work and methodological refinements. In particular, the biochemical validation experiments demonstrating PFOR-dependent H₂ production and the hemin-dependent physiological studies represent valuable additions that strengthen the mechanistic claims.

We thank the reviewer for their excellent suggestions that have substantially enhanced the manuscript.

However, one fundamental issue could be better addressed. While the authors conducted additional experiments in response to my concerns about quantitative significance, they failed to provide clear stoichiometric analysis and thus whether H₂ represents a major or minor metabolic flux in *Bacteroides* remains unclear. Figure S9 shows glucose consumption and H₂ production on different scales and units (mg/100ml glucose vs. ppm H₂), making their claim "H₂ production is approximately proportional to glucose consumption and growth" impossible to evaluate. A simple calculation showing "H₂ accounts for X% of carbon flux" would directly address this question. Without this quantitative framework, readers cannot assess whether the group B [FeFe]-hydrogenase represents a metabolically significant pathway or a minor side reaction, undermining one of the paper's central claims about the importance of this enzyme in gut metabolism.

This is an excellent suggestion and we assume the reviewer refers to electron, not carbon, flux through H₂. We have now calculated the mass balance for *Bacteroides fragilis* fermentation resulting in the new Table S8. This confirmed that H₂ is consistently a major electron sink, accounting for approximately 10% of electrons. This is now described accordingly in the revised manuscript:

L345-348: "Nonetheless, substantial amounts of H₂ was still produced from glucose consumption even in the presence of hemin (Fig. 4e), accounting for 9.9 ± 0.6% of the electrons released during fermentation (Table S8). This suggests that H₂ production is a core part of *Bacteroides* metabolism."

Reviewer #2 (Remarks to the Author):

The authors have adequately addressed my concerns.

We thank the reviewer for their supportive and helpful review.

Reviewer #3 (Remarks to the Author):

The authors have taken my comments and criticisms into account in an excellent manner. I also believe that the criticisms raised by the other reviewers have been adequately addressed. With regard to the terminology used in the manuscript, I recommend checking the text again for consistency. For example, genes encode proteins (in most cases), but they are not themselves encoded, as mentioned, e.g., in lines 135 – 136.

Further suggested changes in this regard:

Line 128: "...metatranscriptomes confirm that these hydrogenase genes are highly..."

Line 370: "...lactate dehydrogenase gene varied..."

Line 375: "...during fermentative growth, the corresponding gene remains transcribed ..."

We thank the reviewer for their support and attention to detail. We have carefully checked the manuscript to ensure consistency with gene vs enzyme terminology, including in the three lines mentioned.